# FeRA: Frequency–Energy Constrained Routing for Effective Diffusion Adaptation Fine-Tuning

**Bo Yin**[1][*]  **Xiaobin Hu**[1][*]  **Xingyu Zhou**[2]  **Yu He**[3]  **Peng-Tao Jiang**[4]  **Yue Liao**[1]  **Junwei Zhu**[5]
**Jiangning Zhang**[6]  **Ying Tai**[7]  **Shuicheng Yan**[1]

## Abstract

Diffusion models have achieved remarkable success in generative modeling, yet how to effectively adapting large pretrained models to new tasks remains challenging. We revisit the reconstruction behavior of diffusion models during denoising to unveil the underlying frequency–energy mechanism governing this process. Building upon this observation, we propose **FeRA**, a frequency-driven fine-tuning framework that aligns parameter updates with the intrinsic frequency–energy progression of diffusion. FeRA establishes a comprehensive frequency–energy framework for effective diffusion adaptation fine-tuning, comprising three synergistic components: *(i)* a compact frequency–energy indicator that characterizes the latent's bandwise energy distribution, *(ii)* a soft frequency router that adaptively fuses multiple frequency-specific adapter experts, and *(iii)* a frequency–energy consistency regularization that stabilizes diffusion optimization and ensures coherent adaptation across bands. Routing operates in both training and inference, with inference-time routing dynamically determined by the latent frequency energy. It integrates seamlessly with adapter-based tuning schemes and generalizes well across diffusion backbones and resolutions. By aligning adaptation with the frequency–energy mechanism, **FeRA** provides a simple, stable, and compatible paradigm for effective and robust diffusion model adaptation. The code are available: https://github.com/YinBo0927/FeRA.git.

---

[*]Equal contribution  [1]National University of Singapore [2]University of Electronic Science and Technology of China [3]Nanyang Technological University [4]vivo [5]Tencent [6]Zhejiang University [7]Nanjing University. Correspondence to: Xiaobin Hu <ben0xiaobin0hu1@nus.edu.sg>.

*Proceedings of the 43^{rd} International Conference on Machine Learning*, Seoul, South Korea. PMLR 306, 2026. Copyright 2026 by the author(s).

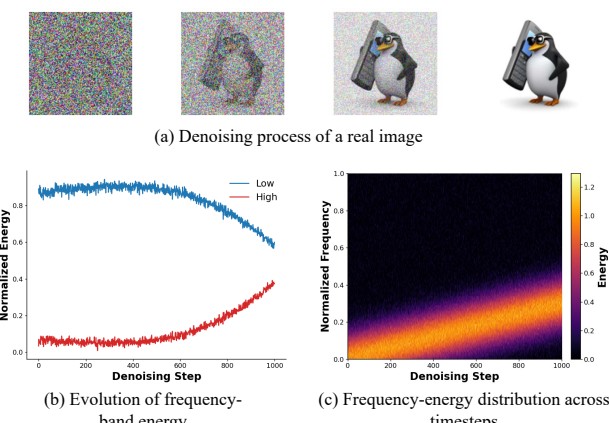

(a) Denoising process of a real image

(b) Evolution of frequency-band energy

(c) Frequency-energy distribution across timesteps

*Figure 1.* Frequency-energy evolution during denoising. (a) Visualization of the denoising process. (b) Evolution of frequency-band energies. (c) Frequency-energy distribution across timesteps.

## 1. Introduction

Diffusion models have fundamentally reshaped generative modeling (Ho et al., 2020; Rombach et al., 2022; Hu et al., 2020; Ji et al., 2025). Advances in denoising-based likelihood learning, consistency training, and distillation have stabilized optimization and accelerated sampling (Ho et al., 2020; Song et al., 2023; Salimans & Ho, 2022; Ji et al., 2024; Xiaobin et al., 2025), while the shift from compact U-Nets to cross-scale attention with strong semantic encoders has improved resolution and alignment (Rombach et al., 2022; Radford et al., 2021). Beyond generic image synthesis, diffusion models now power personalization, editing and controllable generation (Ruiz et al., 2023; Gal et al., 2022; Zhang et al., 2023a; Poole et al., 2022). Open frameworks such as Stable Diffusion (Rombach et al., 2022) bridge research and deployment, making it feasible and increasingly essential to adapt large pretrained backbones to new tasks without harming generalization. This motivates a central challenge: how to efficiently and reliably adapt powerful diffusion models to diverse downstream scenarios within limited computational and storage budgets, placing parameter-efficient fine-tuning (PEFT) (Hu et al., 2022; Mou et al., 2024) at the center of attention.

The research trajectory of PEFT has generally followed

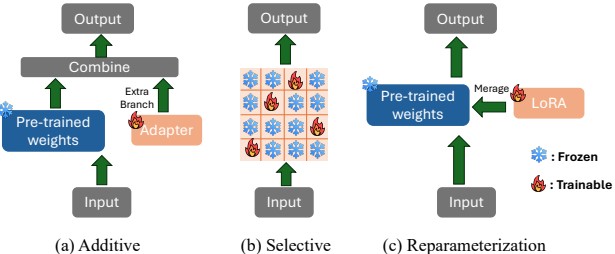

*Figure 2.* The classical parameter-efficient fine-tuning methods.

three main routes in Fig. 2: *1)* **Additive methods** attach external adapters or side branches, which are simple to train but increase inference overhead (Houlsby et al., 2019; Mou et al., 2024). *2)* **Reparameterization methods**, such as low-rank decomposition, embed learnable updates into the original weights for seamless merging at inference (Hu et al., 2022). *3)* **Selective methods** tune only a small subset of parameters or channels, preserving pretrained priors but increasing implementation complexity (Guo et al., 2020). These existing fine-tuning paradigms, including additive, reparameterization, and selective methods, still face inherent limitations. These methods uniformly handle noise across all timesteps. However, diffusion denoising is inherently stage-varying, where the model exhibits distinct noise-signal characteristics over time (Ho et al., 2020; Nichol & Dhariwal, 2021; Karras et al., 2022; Chen et al., 2024; Yang et al., 2023), as shown in Fig. 1. Consequently, such uniform adaptation fails to align with the intrinsic noise-dependent dynamics of the diffusion process. This raises a fundamental question: *Can we design a fine-tuning strategy that adapts to the varying noise conditions throughout the diffusion process rather than enforcing uniform updates across all timesteps?*

Given the stage-varying nature of diffusion denoising, a dynamic adaptation mechanism across timesteps naturally becomes desirable. Such dynamic capacity can, in principle, be achieved through mixture-of-experts (MoE) mechanisms. Inspired by the success of MoE architectures, recent studies attempt to address this limitation by introducing dynamic routing mechanisms that selectively activate different experts across timesteps or layers. Such designs provide adaptive capacity and improve specialization, yet most routing keys remain discrete and are typically driven by timestep or structural index **ignoring the physical prior in diffusion process** that often leads to unstable optimization and limited generalization across diffusion backbones. To move beyond these discrete and task-specific schemes, we explore *whether the routing process itself can be guided by the continuous pattern observed in diffusion denoising rather than by manually defined timestep indices, thereby enabling fine-tuning to follow the model's inherent denoising progression.*

Therefore, we propose **FeRA**, a frequency-energy-driven

framework for parameter-efficient fine-tuning. FeRA employs frequency experts that reflect the frequency-energy progression observed in diffusion denoising. At each step, the latent representation is decomposed into distinct frequency bands, and their energy proportions are fed into a frequency router to produce continuous routing weights. These weights softly blend the frequency experts, enabling the model to adapt smoothly across energy domains while preserving pretrained priors. To further stabilize training, a frequency-consistency regularization constrains the discrepancy of update magnitudes between adjacent bands. The unified routing and regularization generalize across model scales and configurations, providing a stable, transferable, and lightweight paradigm for energy-aware fine-tuning. The main contributions of this work are as follows:

- **Frequency-energy analysis of diffusion denoising.** We reveal a consistent coarse-to-fine progression that links denoising steps to frequency-energy composition.

- **Frequency-driven routing architecture (FeRA).** We propose a Frequency Energy Indicator to characterize the frequency-energy evolution across timesteps, and leverage it to guide a soft frequency-based expert routing mechanism that replaces discrete timestep hard routing.

- **Frequency-energy consistency regularization.** We introduce a lightweight frequency-energy-based regularization that stabilizes training and enhances the transferability of efficient diffusion adaptation.

- **Comprehensive experimental validation.** We conduct extensive experiments across diverse datasets and diffusion backbones, demonstrating consistent improvements in generation quality and generalization, validating the effectiveness of our frequency-driven design.

## 2. Related Work

### 2.1. Diffusion Models

Diffusion models (Ho et al., 2020; Song et al., 2020; Nichol & Dhariwal, 2021) have achieved remarkable success in generative modeling by progressively denoising Gaussian noise into structured images. Latent diffusion (Rombach et al., 2022) further improves efficiency by operating in a compressed latent space, enabling large-scale text-to-image generation exemplified by Stable Diffusion. Subsequent research has extended diffusion models to a wide range of applications, including conditional generation (Zhang et al., 2023a; Mou et al., 2024; Ye et al., 2023), controllable synthesis (Song et al., 2025b;a), video generation (Blattmann et al., 2023; Guo et al., 2023), 3D content creation (Poole et al., 2022; Wang et al., 2023), and audio-visual modeling (Singer

et al., 2022; Popov et al., 2021). Beyond applications, several studies investigate the theoretical and structural properties of diffusion processes, such as noise scheduling (Karras et al., 2022; San-Roman et al., 2021), consistency training (Song et al., 2023), and frequency-domain analysis (Li et al., 2023). These works collectively reveal that diffusion denoising proceeds in a coarse-to-fine manner, where low-frequency structures form before high-frequency details, providing new insights for enhancing generative quality and interpretability.

## 2.2. PEFT for Generative Models

Parameter-efficient fine-tuning (PEFT) techniques have become essential for adapting large diffusion models to new concepts or domains without full retraining (Liu et al., 2022; Cao et al., 2025). Early approaches such as Textual Inversion (Gal et al., 2022) and DreamBooth (Ruiz et al., 2023) personalize models by learning small embeddings or tuning only a subset of parameters. LoRA (Hu et al., 2022) introduces low-rank adaptation and has been widely adopted in diffusion frameworks for its efficiency and scalability (Zhang et al., 2023b). Extensions including ControlNet (Zhang et al., 2023a), T2I-Adapter (Mou et al., 2024), and IP-Adapter (Ye et al., 2023) integrate auxiliary networks to enhance conditional control, while adapter fusion and expert routing (Li et al., 2025; Zhu et al., 2024) further improve flexibility and robustness. More recent studies investigate unified and task-aware PEFT architectures (Hu et al., 2025; Yin et al., 2025), reflecting a growing interest in scalable and modular adaptation for large generative models. Despite differences in structure and application, these methods share a common objective: to enable controllable and data-efficient fine-tuning, yet they implicitly assume isotropic denoising dynamics and overlook the timestep-wise anisotropy of diffusion reconstruction.

## 2.3. Mixture of Expert

Mixture-of-Experts (MoE) architectures introduce conditional computation by activating different parameter subsets based on input features or contextual cues (Shazeer et al., 2017; Fedus et al., 2022; Valadarsky et al., 2017). This paradigm improves model capacity and specialization without linearly increasing inference cost. Recent works extend MoE to vision and generative tasks, where expert routing is driven by spatial location, semantic content, or timestep signals (Ganjdanesh et al., 2024). In diffusion models, MoE-based adapters or routers have been explored to decouple timestep-dependent behaviors (Liu et al., 2024b; Zhu et al., 2024; Park et al., 2023; Lee et al., 2024; Park et al., 2024; Fei et al., 2024), allowing distinct experts to handle different denoising stages. However, most of these methods rely on discrete timestep gating, which introduces hard boundaries and unstable expert activation during training.

## 3. Frequency-Energy in Denoising

A diffusion model gradually transforms Gaussian noise into a structured image through iterative denoising (Ho et al., 2020; Yu et al., 2025; Tivnan et al., 2025; Xu et al., 2020). As illustrated in Fig. 1(a), this process exhibits a clear visual transition from chaotic noise to coherent structure. To analyze this progression quantitatively, we compute the 2D Fourier amplitude spectrum $A_t(f)$ of each intermediate image $x_t$. By integrating spectral energy within predefined low- and high-frequency bands, we obtain their normalized evolution over timesteps (Fig. 1(b)). The results reveal a consistent shift of energy dominance from low to high frequencies as denoising proceeds. The frequency-energy distribution in Fig. 1(c) further confirms that high-frequency components only become prominent in later stages, indicating that diffusion reconstruction progressively shifts energy from low to high frequencies

This behavior can be explained through the frequency-dependent signal-to-noise ratio (SNR) of the diffusion process (Arora et al., 2024). Given the forward formulation $x_t = \sqrt{\alpha_t}\, x_0 + \sqrt{1 - \alpha_t}\, \epsilon$, its Fourier-domain representation is

$$\hat{x}_t(f) = \sqrt{\alpha_t}\, \hat{x}_0(f) + \sqrt{1 - \alpha_t}\, \hat{\epsilon}(f). \tag{1}$$

The per-frequency SNR is defined as

$$\mathrm{SNR}_t(f) = \frac{\alpha_t |\hat{x}_0(f)|^2}{(1 - \alpha_t)\, \mathbb{E}[|\hat{\epsilon}(f)|^2]} \;\propto\; \frac{\alpha_t}{(1 - \alpha_t) f^\gamma}, \tag{2}$$

since natural images approximately follow a power-law spectrum $|\hat{x}_0(f)|^2 \propto 1/f^\gamma$ with $\gamma \approx 2$ (Field, 1987; Ruderman & Bialek, 1993), implying that low frequencies carry substantially higher energy than high frequencies and we note that the denominator $\mathbb{E}[|\hat{\epsilon}(f)|^2]$ remains nearly constant across frequencies, so the frequency dependence of $\mathrm{SNR}_t(f)$ primarily arises from the signal term. This relation indicates that both smaller $\alpha_t$ and higher $f$ lead to rapidly diminishing SNR. When $\alpha_t$ is small at early timesteps, the signal term $|\hat{x}_0(f)|^2$ becomes negligible compared to the noise power. This effect is particularly pronounced at high frequencies, where the intrinsic decay $|\hat{x}_0(f)|^2 \propto 1/f^\gamma$ further suppresses the signal energy. As a result, the observed spectrum $\hat{x}_t(f)$ in these bands is effectively noise-dominated and contains almost no recoverable structure. In contrast, low-frequency components maintain substantially higher SNR, allowing coarse spatial layouts to remain statistically discernible even in the early stages of denoising. As $\alpha_t$ increases over time, the effective SNR of high-frequency bands gradually improves, enabling the model to recover fine-grained details in later denoising steps.

Since the VAE encoder $\mathcal{E}$ is locally approximately linear within the manifold of natural images (Bengio et al., 2013;

Kingma et al., 2019; Rombach et al., 2022), its latent representation $z_t = \mathcal{E}(x_t)$ satisfies $\hat{z}_t(f) \approx H(f)\hat{x}_t(f)$, where $H(f)$ denotes the encoder's local frequency response that approximately preserves the spectral structure of natural images, leading to a matching latent SNR, $\mathrm{SNR}_t^{(z)}(f) \approx \mathrm{SNR}_t(f)$. Hence, the same evolution persists within the latent domain, where we later define our frequency indicators and routing strategy, ensuring that the model's adaptive behavior remains aligned with the underlying spectral progression throughout the entire denoising trajectory.

## 4. Frequency-Energy Constrained Routing

Motivated by the frequency-energy analysis in Section 3, we propose **FeRA**, a parameter-efficient fine-tuning framework that introduces frequency-energy-aware mechanisms at both architectural and training levels. As illustrated in Fig. 3, FeRA first employs a Difference-of-Gaussians (DoG) (Lowe, 2004) operator to extract the relative energy distribution across different frequency bands, forming a compact *Frequency-Energy Indicator (FEI)* that characterizes the latent's frequency-energy state. The FEI is then fed into a *Soft Frequency Router*, which adaptively blends multiple LoRA experts according to the current frequency-energy composition, providing a continuous and interpretable alternative to timestep-based hard routing. To further ensure consistency during training, FeRA incorporates a *Frequency-Energy Consistency Loss (FECL)* that aligns the denoising trajectory with the inherent frequency-energy evolution. We also give the theoretical analysis of frequency-energy routing compared with timestep routing in Appendix B and discuss the potential applicability of our method to non-natural images in the Appendix C.

### 4.1. Frequency-Energy Indicator (FEI)

We define a simple descriptor that summarizes the frequency-energy of the latent feature $z_t \in \mathbb{R}^{C \times H \times W}$ at denoising step $t$. Let $G_\sigma$ be a Gaussian blur with standard deviation $\sigma$ measured in latent pixels. In the frequency domain with radial frequency $\rho$, its response is $\widehat{G}_\sigma(\rho) = e^{-2\pi^2 \sigma^2 \rho^2}$. A Difference-of-Gaussians (DoG) $D_{\sigma_i, \sigma_j} = G_{\sigma_i} - G_{\sigma_j}$ is therefore a band-pass filter with response $e^{-2\pi^2 \sigma_i^2 \rho^2} - e^{-2\pi^2 \sigma_j^2 \rho^2}$, which is small near $\rho = 0$ and at high $\rho$, and peaks at a middle range. Using $n$ Gaussian kernels with scales $\{\sigma_1, \ldots, \sigma_n\}$ in a geometric progression (we use $\sigma_k = \kappa \cdot 2^{k-1}$ for $k = 1, \ldots, n$, with $\kappa = \min(H, W)/128$, implicitly corresponding to increasing frequency cutoffs), we construct $n$ frequency bands. The filtered components are computed as follows:

$$z_t^{(k)} = \begin{cases} G_{\sigma_k} * z_t & \text{if } k = 1, \\ (G_{\sigma_k} - G_{\sigma_{k-1}}) * z_t & \text{if } 1 < k < n, \\ (\mathbb{I} - G_{\sigma_{n-1}}) * z_t & \text{if } k = n, \end{cases} \quad (3)$$

where $G_{\sigma_k}$ denotes a Gaussian blur with standard deviation $\sigma_k$, and $*$ is depth-wise convolution. We measure the energy of each band as:

$$E_t^{(k)} = \|z_t^{(k)}\|_2^2 = \sum_{c=1}^{C} \sum_{x=1}^{H} \sum_{y=1}^{W} \left( z_{t,c}^{(k)}(x,y) \right)^2, \quad k = 1, \ldots, n. \quad (4)$$

By Parseval's theorem (Kwakernaak & Sivan, 1991), the spatial-domain per-band energy $E_t^{(k)}$ equals its frequency-domain counterpart. Hence, the computed per-band quantity $E_t^{(k)}$ is exactly the frequency-domain energy of $z_t^{(k)}$. With the dyadic spacing above, the filters exhibit minimal overlap and cover the full frequency spectrum, such that $\sum_{k=1}^{n} E_t^{(k)} \approx \|z_t\|_2^2$. We define the **Frequency-Energy Indicator (FEI)** as the normalized energy vector:

$$\mathbf{e}_t = \frac{[E_t^{(1)}, \ldots, E_t^{(n)}]^\top}{\sum_{k=1}^{n} E_t^{(k)}} \in \mathbb{R}^n, \quad (5)$$

Which lies on the probability simplex and is invariant to global rescaling of $z_t$, offering a stable descriptor. In Appendix A, we further analysis that FEI is not a proxy of timestep.

### 4.2. Soft Frequency Router

Building upon FEI, we introduce a **Frequency-Aware Routing** mechanism that dynamically blends multiple LoRA experts according to the latent's spectral state. Specifically, the FEI is projected through a lightweight MLP router $g_\phi$ to produce $M$ routing logits and weights:

$$\boldsymbol{\alpha}_t = \mathrm{softmax}\left( \frac{g_\phi(\mathbf{e}_t)}{\tau} \right) \in \mathbb{R}^M \quad (6)$$

where $\mathbf{e}_t$ is FEI, $\tau$ controls routing softness, and $\alpha_{t,m}$ is the weight assigned to expert $\mathcal{E}_k$. The routed adapter output is the weighted mixture

$$y_t = \sum_{m=1}^{M} \alpha_{t,m} \, \mathcal{E}_m(z_t) \quad (7)$$

enabling smooth transitions across different frequency experts as the energy distribution evolves. This frequency-driven formulation improves continuity and interpretability over timestep-based hard gating. In practice we attach the frequency-energy experts to the same layers, use a small two-layer router, and fix $\tau = 0.7$.

### 4.3. Frequency-Energy Consistency Loss (FECL)

While the frequency router adaptively adjusts the contribution of LoRA experts, it does not explicitly regularize the spectral behavior of the latent representation. To enforce a consistent evolution of frequency energy during denoising, we introduce a **Frequency-Energy Consistency Loss (FECL)** applied in the latent space.

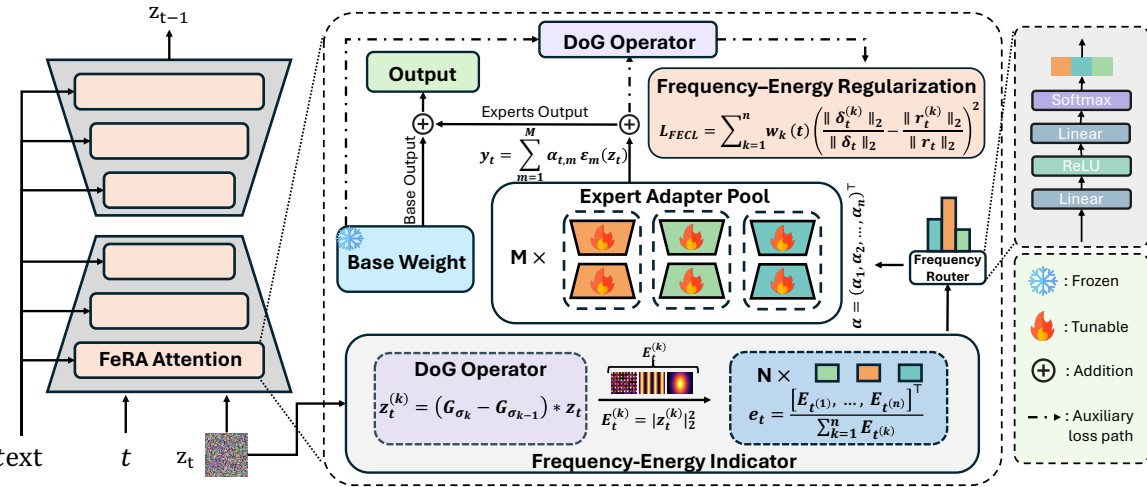

*Figure 3.* Overview of the FeRA framework. The **Frequency-Energy Indicator (FEI)** extracted by DoG operators guides a **Soft Frequency Router** to adaptively blend multiple LoRA experts. A **Frequency-Energy Consistency Loss (FECL)** further regularizes the spectral alignment between correction and residual during fine-tuning.

Let the latent predicted by the base model be $z_t^{\text{base}}$ and the one adapted by LoRA be $z_t^{\text{lora}}$. We define the correction and reconstruction errors as

$$\delta_t = z_t^{\text{lora}} - z_t^{\text{base}}, \qquad r_t = z_t^{\text{lora}} - z_t, \qquad (8)$$

where $z_t$ is the ground-truth latent at step $t$. Using the $n$-band decomposition from Sec. 4.1, we apply DoG filters $D_{\sigma_{k-1},\sigma_k}$ to obtain bandwise components:

$$\left(\delta_t^{(1)}, \dots, \delta_t^{(n)}\right) = D(\delta_t), \quad \left(r_t^{(1)}, \dots, r_t^{(n)}\right) = D(r_t), \qquad (9)$$

where $\delta_t = z_t^{\text{lora}} - z_t^{\text{base}}$ and $r_t = z_t^{\text{lora}} - z_t$ denote the adapter correction and residual error, respectively. We then align the correction with the residual in the frequency domain via the **Frequency-Energy Consistency Loss (FECL)**:

$$\mathcal{L}_{\text{FECL}} = \sum_{k=1}^{n} w_k(t) \left( \frac{\|\delta_t^{(k)}\|_2}{\|\delta_t\|_2} - \frac{\|r_t^{(k)}\|_2}{\|r_t\|_2} \right)^2 \qquad (10)$$

where $w_k(t)$ are frequency-band weights derived from the current FEI (we use $w_k(t) = \tilde{E}_t^{(k)} / \sum_j \tilde{E}_t^{(j)}$). The full objective is

$$\mathcal{L} = \mathcal{L}_{\text{denoise}} + \lambda_{\text{f}} \mathcal{L}_{\text{FECL}}. \qquad (11)$$

Minimizing $\mathcal{L}_{\text{FECL}}$ enforces *frequency-energy alignment* between the adapter correction and the residual across frequencies, concentrating updates where residual energy is present and suppressing updates where it is negligible.

## 5. Experiment

To validate the effectiveness of our method, we conduct experiments on two representative scenarios: downstream dataset fine-tuning and image customization. Specially, we set the number of FEI and LoRA experts to 3, and why

we set this number will be discussed in Section 5.3. We compare our approach with four state-of-the-art parameter-efficient fine-tuning methods: LoRA (Hu et al., 2022), DoRA (Liu et al., 2024a) AdaLoRA (Zhang et al., 2023b), and SaRA (Hu et al., 2025), along with the full-parameter fine-tuning baseline. All models are fine-tuned under same training configurations for comparison.

### 5.1. Text-to-Image Style Adaptation

**Experiment Setting.** In this experiment, we fine-tune on five widely used CIVITAI style datasets, including Barbie Style, Cyberpunk Style, Elementfire Style, Expedition Style, and Hornify Style. We compare parameter-efficient fine-tuning methods on Stable Diffusion 2.0, 3.0, and FLUX.1 under three trainable parameter budgets of 5M, 20M, and 50M, scaling the rank of FeRA inversely to the number of experts to maintain a fair iso-parameter comparison with baselines. More backbones can be found in Appendix. Results on additional backbones following the same protocol are provided in the Appendix. Each fine-tuned model is evaluated using three metrics: Fréchet Inception Distance (FID), CLIP Score, and Style Score for perceptual style consistency. The Style Score is obtained through a multi-modal large language model (MLLM)-based quantitative evaluation using Qwen2.5-VL-7B-Instruct (Bai et al., 2023). We also give user study analysis in Appendix F.

**Result Analysis.** The quantitative results are reported in Tab. 1. *1).* Across SD 2.0, 3.0 and FLUX1., FeRA consistently ranks among the top-performing methods across most styles, achieving lower FID while maintaining competitive CLIP and Style scores. Even under the 5M budget, it attains the lowest FID on several datasets without compromising alignment. *2).* As the trainable budget increases to 20M and 50M, FeRA captures finer stylistic details, with FID

*Table 1.* Comparison with different parameter-efficient fine-tuning methods on Stable Diffusion 2.0, 3.0 and FLUX.1. **Orange bold** = best *Light orange italic* = second best. ***Notice:*** **the training parameter is same for fair comparison by controlling the rank of FeRA.**

| Backbone | Params | Method | Barbie CLIP↑ | FID↓ | Style↑ | Cyberpunk CLIP↑ | FID↓ | Style↑ | Expedition CLIP↑ | FID↓ | Style↑ | Hornify CLIP↑ | FID↓ | Style↑ | Elementfire CLIP↑ | FID↓ | Style↑ |
|---|---|---|---|---|---|---|---|---|---|---|---|---|---|---|---|---|---|
| SD 2.0 | 5M | LoRA | 35.67 | 213.42 | 8.46 | 33.23 | 178.17 | 8.33 | 31.85 | 164.21 | 8.54 | 32.31 | 175.16 | 8.57 | 32.18 | 200.45 | 8.60 |
| | | DoRA | 35.61 | 211.63 | 8.49 | 33.26 | 176.48 | 8.34 | 31.86 | 162.93 | 8.55 | 32.33 | 173.12 | 8.58 | 32.20 | 198.55 | 8.61 |
| | | AdaLoRA | 35.61 | 212.45 | 8.48 | 33.26 | 177.30 | 8.37 | 31.86 | 163.61 | 8.56 | 32.34 | 174.12 | 8.59 | 32.12 | 199.50 | 8.62 |
| | | SaRA | 35.53 | 207.12 | 8.51 | 33.28 | 170.81 | 8.36 | 31.88 | 160.37 | 8.58 | 32.26 | 170.28 | 8.60 | 32.36 | 190.28 | 8.64 |
| | | **FeRA (Ours)** | 36.63 | 201.94 | 8.52 | 33.38 | 166.54 | 8.39 | 33.98 | 156.36 | 8.61 | 32.46 | 166.02 | 8.63 | 33.32 | 189.94 | 8.65 |
| | 20M | LoRA | 35.78 | 198.95 | 8.74 | 33.35 | 165.17 | 8.45 | 31.97 | 152.17 | 7.57 | 32.44 | 162.91 | 8.53 | 32.30 | 186.24 | 8.22 |
| | | DoRA | 35.72 | 197.12 | 8.15 | 33.38 | 164.31 | 8.52 | 31.98 | 151.52 | 8.03 | 32.46 | 160.95 | 8.15 | 32.24 | 184.46 | 8.62 |
| | | AdaLoRA | 35.72 | 197.49 | 8.31 | 33.38 | 164.18 | 8.76 | 31.98 | 152.12 | 8.17 | 32.46 | 161.88 | 8.56 | 32.24 | 185.52 | 8.64 |
| | | SaRA | 35.65 | 192.36 | 8.44 | 33.40 | 158.48 | 8.34 | 32.00 | 149.13 | 8.54 | 32.39 | 158.24 | 7.99 | 32.48 | 177.09 | 8.76 |
| | | **FeRA (Ours)** | 35.74 | 187.83 | 8.82 | 33.50 | 154.19 | 8.78 | 32.12 | 145.64 | 8.55 | 32.58 | 154.14 | 8.89 | 33.44 | 178.51 | 8.87 |
| | 50M | LoRA | 35.80 | 200.12 | 8.16 | 33.37 | 166.83 | 8.73 | 31.98 | 151.92 | 8.27 | 32.45 | 162.15 | 8.19 | 32.31 | 185.74 | 8.44 |
| | | DoRA | 35.74 | 198.33 | 8.00 | 33.40 | 165.24 | 7.83 | 31.99 | 150.47 | 8.67 | 32.47 | 160.21 | 8.16 | 32.25 | 183.96 | 8.06 |
| | | AdaLoRA | 35.74 | 199.31 | 8.95 | 33.40 | 165.91 | 8.18 | 31.99 | 151.13 | 8.35 | 32.47 | 161.00 | 8.14 | 32.25 | 184.84 | 8.41 |
| | | SaRA | 35.67 | 193.18 | 8.47 | 33.42 | 160.03 | 7.92 | 32.01 | 148.33 | 8.71 | 32.40 | 157.24 | 8.55 | 32.49 | 176.25 | 7.86 |
| | | **FeRA (Ours)** | 36.76 | 189.20 | 8.98 | 33.52 | 156.31 | 8.75 | 32.31 | 144.65 | 8.83 | 32.59 | 153.36 | 9.06 | 33.45 | 175.72 | 8.77 |
| | 866M | Full-Tuning | 35.70 | 191.00 | 8.95 | 33.45 | 158.60 | 8.72 | 32.25 | 146.20 | 8.80 | 32.52 | 155.10 | 9.02 | 32.38 | 179.90 | 8.74 |
| SD 3.0 | 5M | LoRA | 36.05 | 189.08 | 8.41 | 33.67 | 163.46 | 8.30 | 31.61 | 147.65 | 8.49 | 32.21 | 170.31 | 8.53 | 32.19 | 170.39 | 8.54 |
| | | DoRA | 36.09 | 189.62 | 8.43 | 33.72 | 158.02 | 8.29 | 31.60 | 147.18 | 8.52 | 32.22 | 172.16 | 8.55 | 32.20 | 169.27 | 8.56 |
| | | AdaLoRA | 36.08 | 188.91 | 8.44 | 33.72 | 161.51 | 8.31 | 31.61 | 147.43 | 8.54 | 32.22 | 170.12 | 8.55 | 32.22 | 169.80 | 8.57 |
| | | SaRA | 36.00 | 186.50 | 8.45 | 33.71 | 155.22 | 8.28 | 31.64 | 141.02 | 8.55 | 33.15 | 171.27 | 8.56 | 32.25 | 170.32 | 8.58 |
| | | **FeRA (Ours)** | 36.09 | 184.09 | 8.48 | 33.81 | 154.51 | 8.33 | 31.67 | 139.05 | 8.57 | 32.34 | 174.25 | 8.59 | 31.79 | 168.23 | 8.60 |
| | 20M | LoRA | 36.15 | 176.41 | 8.55 | 33.78 | 152.41 | 7.81 | 31.72 | 137.34 | 8.59 | 32.33 | 158.35 | 8.27 | 32.30 | 158.93 | 8.36 |
| | | DoRA | 36.20 | 176.92 | 8.77 | 33.83 | 147.55 | 8.02 | 31.71 | 136.69 | 7.86 | 32.34 | 160.22 | 8.30 | 32.31 | 157.84 | 8.26 |
| | | AdaLoRA | 36.19 | 176.24 | 8.76 | 33.83 | 150.27 | 7.90 | 31.71 | 137.12 | 8.52 | 32.34 | 158.42 | 8.57 | 32.31 | 158.31 | 8.67 |
| | | SaRA | 36.10 | 173.91 | 8.23 | 33.82 | 144.59 | 7.79 | 31.75 | 131.20 | 7.64 | 33.27 | 159.13 | 8.61 | 32.36 | 154.31 | 7.79 |
| | | **FeRA (Ours)** | 37.20 | 171.55 | 8.78 | 33.92 | 141.22 | 8.54 | 31.78 | 129.43 | 8.60 | 32.46 | 151.94 | 8.83 | 31.90 | 150.86 | 8.71 |
| | 50M | LoRA | 36.18 | 177.48 | 8.73 | 33.80 | 153.13 | 7.99 | 31.73 | 136.72 | 8.47 | 32.35 | 157.87 | 7.87 | 32.32 | 158.71 | 8.26 |
| | | DoRA | 36.23 | 178.63 | 8.01 | 33.85 | 148.74 | 7.68 | 31.72 | 136.24 | 8.59 | 32.36 | 159.64 | 8.39 | 32.33 | 157.53 | 7.61 |
| | | AdaLoRA | 36.22 | 177.64 | 8.14 | 33.85 | 151.95 | 7.49 | 31.72 | 136.54 | 8.08 | 32.36 | 157.53 | 7.91 | 32.33 | 157.54 | 8.52 |
| | | SaRA | 36.13 | 175.13 | 8.55 | 33.42 | 160.03 | 7.92 | 32.01 | 148.33 | 8.71 | 32.40 | 157.24 | 8.55 | 32.38 | 158.12 | 8.13 |
| | | **FeRA (Ours)** | 37.23 | 172.59 | 8.81 | 33.94 | 145.42 | 8.42 | 32.79 | 128.73 | 8.65 | 33.48 | 161.12 | 8.76 | 31.92 | 154.95 | 8.57 |
| | 2085M | Full-Tuning | 36.15 | 174.80 | 8.78 | 33.85 | 147.90 | 8.39 | 31.72 | 130.50 | 8.62 | 32.40 | 162.80 | 8.72 | 31.85 | 156.90 | 8.54 |
| FLUX.1 | 5M | LoRA | 36.05 | 175.23 | 8.55 | 33.61 | 146.32 | 8.34 | 32.08 | 132.46 | 8.19 | 32.65 | 143.51 | 8.66 | 32.52 | 159.35 | 7.82 |
| | | DoRA | 36.01 | 173.58 | 8.51 | 33.64 | 144.78 | 8.28 | 32.10 | 131.34 | 8.63 | 32.67 | 141.83 | 8.14 | 32.75 | 157.18 | 7.77 |
| | | AdaLoRA | 36.03 | 172.56 | 8.52 | 33.65 | 143.51 | 8.15 | 31.98 | 132.22 | 8.38 | 32.66 | 140.30 | 7.95 | 32.49 | 155.98 | 8.01 |
| | | SaRA | 36.08 | 170.12 | 8.61 | 33.66 | 141.59 | 7.93 | 32.12 | 129.42 | 8.45 | 32.70 | 139.23 | 7.87 | 32.58 | 154.37 | 8.10 |
| | | **FeRA (Ours)** | 37.05 | 165.87 | 8.66 | 33.76 | 138.05 | 8.50 | 32.02 | 126.18 | 8.72 | 33.81 | 135.75 | 8.75 | 33.68 | 155.51 | 8.77 |
| | 20M | LoRA | 36.15 | 162.79 | 8.35 | 33.74 | 136.43 | 8.45 | 32.20 | 122.19 | 8.78 | 32.78 | 133.40 | 8.89 | 32.65 | 148.53 | 8.33 |
| | | DoRA | 36.12 | 158.92 | 8.86 | 33.77 | 134.34 | 8.36 | 32.21 | 121.77 | 8.43 | 32.85 | 128.05 | 8.23 | 32.68 | 146.44 | 8.81 |
| | | AdaLoRA | 36.12 | 161.46 | 8.55 | 33.77 | 135.16 | 8.08 | 32.23 | 122.73 | 8.60 | 32.83 | 132.63 | 8.20 | 32.68 | 147.62 | 8.22 |
| | | SaRA | 36.19 | 156.15 | 8.28 | 33.79 | 131.42 | 8.21 | 32.25 | 120.34 | 8.50 | 32.83 | 129.13 | 8.30 | 32.71 | 143.29 | 8.39 |
| | | **FeRA (Ours)** | 37.16 | 153.79 | 9.01 | 33.89 | 128.43 | 8.94 | 32.14 | 117.83 | 8.79 | 32.94 | 126.11 | 9.06 | 32.81 | 141.15 | 9.10 |
| | 50M | LoRA | 35.17 | 163.76 | 8.43 | 33.75 | 136.85 | 8.39 | 32.21 | 122.23 | 8.65 | 32.79 | 132.93 | 8.43 | 32.66 | 148.40 | 8.78 |
| | | DoRA | 36.13 | 159.02 | 8.71 | 33.78 | 135.61 | 8.97 | 32.14 | 121.15 | 8.02 | 32.84 | 131.00 | 8.28 | 32.69 | 145.39 | 8.65 |
| | | AdaLoRA | 36.13 | 162.62 | 8.55 | 33.78 | 135.93 | 8.64 | 32.24 | 121.77 | 8.06 | 32.84 | 131.48 | 8.92 | 32.69 | 146.67 | 8.34 |
| | | SaRA | 36.20 | 156.91 | 8.18 | 33.80 | 131.28 | 8.19 | 33.26 | 119.71 | 8.55 | 32.84 | 128.65 | 9.00 | 32.72 | 143.74 | 8.64 |
| | | **FeRA (Ours)** | 37.17 | 154.76 | 9.15 | 34.90 | 129.32 | 9.11 | 32.15 | 117.22 | 8.85 | 33.95 | 125.36 | 9.16 | 33.82 | 141.21 | 9.03 |
| | 7.8B | Full-Tuning | 36.13 | 165.37 | 8.71 | 33.81 | 137.55 | 8.55 | 32.17 | 125.68 | 8.77 | 32.86 | 135.25 | 8.80 | 32.80 | 153.87 | 8.82 |

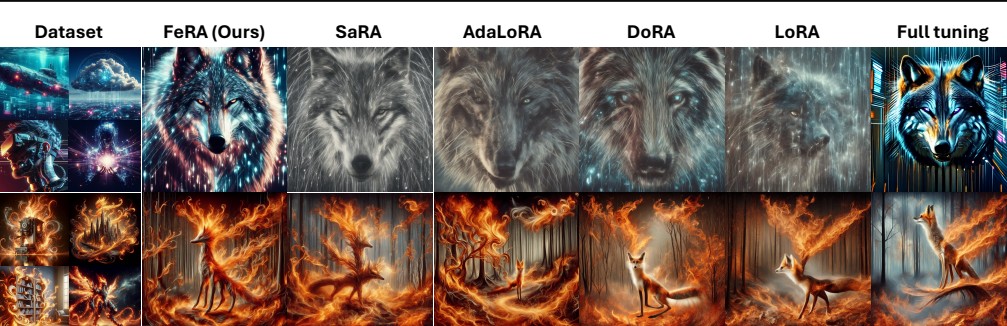

*Figure 4.* Comparison of the generated images between different PEFT methods.

continuing to improve and Style scores remaining stable, indicating that frequency-energy-aware soft routing scales effectively with model capacity rather than overfitting. *3).* The full-parameter baseline is not uniformly dominant under

*Table 2.* Quantitative comparison between different PEFT methods on image customization. **Red bold** = best, *orange italic* = second best.

| Methods | Dog | | Clock | | Backpack | | Toy Duck | | Teapot | |
|---|---|---|---|---|---|---|---|---|---|---|
| | CLIP-I ↑ | CLIP-T ↑ | CLIP-I ↑ | CLIP-T ↑ | CLIP-I ↑ | CLIP-T ↑ | CLIP-I ↑ | CLIP-T ↑ | CLIP-I ↑ | CLIP-T ↑ |
| Dreambooth + Full-tuning | 0.788 | 24.15 | 0.789 | 23.15 | 0.654 | 24.09 | 0.790 | 24.05 | 0.750 | 24.12 |
| Dreambooth + LoRA | *0.895* | 23.64 | 0.913 | 21.71 | *0.917* | 25.23 | 0.905 | 23.80 | 0.906 | 23.58 |
| Dreambooth + DoRA | 0.897 | 23.71 | *0.915* | 21.78 | 0.914 | *25.31* | *0.907* | 23.88 | *0.908* | 23.65 |
| Dreambooth + AdaLoRA | 0.896 | 23.69 | 0.914 | 21.76 | 0.917 | 25.29 | 0.906 | 23.85 | 0.907 | 23.63 |
| Dreambooth + SaRA | 0.790 | **25.97** | 0.887 | *23.51* | 0.886 | 25.27 | 0.885 | **25.50** | 0.866 | *25.12* |
| **Dreambooth + FeRA (Ours)** | **0.900** | *25.95* | **0.920** | **23.63** | **0.925** | **25.35** | **0.910** | *24.60* | **0.913** | **25.32** |

*Figure 5.* DreamBooth results across PEFT methods. FeRA delivers more consistent identity and cleaner compositions.

these settings, while FeRA achieves comparable or better performance with substantially fewer parameters. Qualitative comparisons in Fig. 4 further shows coherent structure, accurate alignment, and high-fidelity style renderings.

## 5.2. Image Customization

**Experiment Setting.** We evaluate the proposed FeRA framework on personalized image generation, a representative task that tests a model's ability to adapt to novel identities or concepts with limited examples. Specifically, we conduct experiments on five representative subject categories, including dog, clock, backpack, toy duck, and teapot. Following the standard DreamBooth protocol (Ruiz et al., 2023), we fine-tune a pre-trained Stable Diffusion 2.0 model using a small number of instance images (3-5) per subject, binding each identity to a unique rare token. All competing methods are implemented on the same UNet backbone for a fair comparison, including full finetuning, LoRA (Hu et al., 2022), DoRA (Liu et al., 2024a), AdaLoRA (Zhang et al., 2023b), and SaRA (Hu et al., 2025). For quantitative evaluation, we use CLIP-based similarity metrics (Ramesh et al., 2022) to measure both image-text alignment and visual fidelity, reporting CLIP-IMG and CLIP-Text scores, where higher values indicate better consistency between generated results and target identity descriptions.

**Result Analysis.** As summarized in Tab. 2, FeRA consistently achieves the highest or second-highest scores on both metrics across all categories. For instance, it attains the best CLIP-I values on all five subjects and top CLIP-T results in four of them, surpassing other PEFT methods by a clear margin. LoRA and DoRA perform competitively in CLIP-I

but yield noticeably lower CLIP-T, suggesting overfitting toward visual features at the expense of text alignment. SaRA, while producing the highest CLIP-T on dog, suffers a sharp drop in CLIP-I, reflecting poor balance between fidelity and semantic accuracy. Overall, FeRA demonstrates the best trade-off between subject identity preservation and text-image alignment, validating its effectiveness and generalizability in personalized image generation. As illustrated in Fig. 5, FeRA produces clearer textures and more accurate colors across all prompts. It preserves structural coherence and fine details, yielding sharper boundaries, more faithful appearance, and more stable rendering across diverse visual conditions, consistently outperforming existing PEFT baselines in visual quality.

## 5.3. Ablation Studies

We conduct ablation studies on Stable Diffusion 2.0 using the Cyberpunk dataset to evaluate FeRA's components: FEI, soft router, FECL, and expert/band counts.

**Timestep-based v.s. Frequency-Energy routing.** Tab. 3 reports that replacing timestep-based routing with FEI consistently improves FID, CLIP, and Style scores. It indicates that *1)* **Frequency–energy outperform timestep-based routing** by providing richer and more informative signals; *2)* **Soft routing outperforms hard routing** due to its smoother and more flexible information aggregation.

**The number of Experts.** Under the three-band setting, Figure 6 (a) compares different numbers of LoRA experts. Using one expert per band (three in total) achieves the best performance across FID, CLIP, and Style metrics. This confirms that frequency-specialized experts enable more

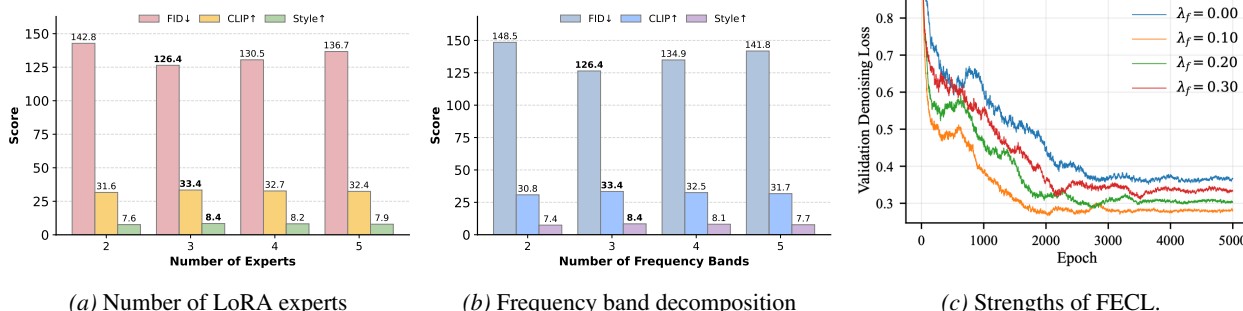

*(a)* Number of LoRA experts      *(b)* Frequency band decomposition      *(c)* Strengths of FECL.

*Figure 6.* Ablation on design factors of FeRA: (a) LoRA expert number, (b) frequency decomposition, and (c) the strengths of FECL.

*Table 3.* Ablation on routing configurations. "FEI" denotes the proposed frequency–energy indicator, and "Soft Router" refers to the soft routing mechanism. Removing FEI falls back to timestep-based routing. "✓" and "✗" indicate whether each module is enabled.

| FEI | Soft Router | CLIP↑ | FID↓ | Style↑ |
|-----|-------------|-------|------|--------|
| ✗ | ✗ | 30.12 | 138.50 | 7.42 |
| ✓ | ✗ | 31.84 | 132.70 | 7.93 |
| ✗ | ✓ | 31.10 | 134.90 | 7.68 |
| ✓ | ✓ | **32.96** | **126.40** | **8.21** |

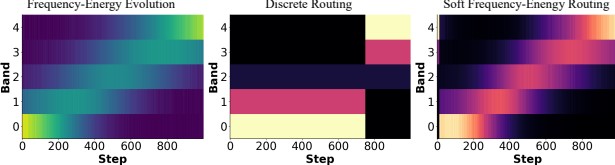

*Figure 7.* Compared with discrete routing strategy.

efficient and interpretable adaptation.

**The number of Frequency Band.** We vary the number of frequency bands used for spectral decomposition in FEI. As shown in Figure 6 (b), three bands (low, mid, high) consistently outperform two or more than three. This indicates that a three-band division captures semantic structure while avoiding redundancy and over-fragmentation.

**Frequency-Energy Consistency Loss.** Figure 6 (c) shows the impact of different FECL weights $\lambda_f \in \{0, 0.1, 0.2, 0.3\}$. A small weight accelerates and stabilizes convergence, whereas removing FECL slows optimization and larger weights cause oscillation. Moderate regularization achieves the best trade-off between stability and generalization.

### 5.4. Our Soft Frequency-Energy Routing Analysis

A representative discrete routing strategy employs a expert MoE design for diffusion denoising: a high-noise expert for early steps that captures global structure, and a low-noise expert for later steps that refines details. The routing is typically governed by a monotonic SNR schedule with a single threshold, resulting in a hard switch between experts. Such a scheme is adopted, for example, in Wan2.2 (Wan et al.,

*Table 4.* Inference-time comparison between LoRA and FeRA across multiple Stable Diffusion backbones.

| Method | SD 1.5 | SD 2.0 | SD 3.0 | SDXL | FLUX.1 |
|--------|--------|--------|--------|------|--------|
| LoRA (baseline) | 1.00× | 1.00× | 1.00× | 1.00× | 1.00× |
| **FeRA (ours)** | 1.08× | 1.08× | 1.07× | 1.05× | 1.04× |

2025). Although this threshold-based routing can be extended to multi-LoRA settings for comparison, it inherently enforces a discrete expert transition. To visualize the behavioral difference, we plot the Frequency-Energy Evolution and the corresponding routing weights as heatmaps (Fig. 7). The discrete routing shows an abrupt switch between low- and high-frequency bands, whereas our frequency-aware routing produces a smooth transition that follows the spectral energy migration. This continuity enables finer control and better alignment between the model's update behavior and the evolving frequency composition during denoising.

### 5.5. Inference-Time Comparison

To quantify inference overhead, we compare LoRA and FeRA using the end-to-end wall-clock latency per generated image across different backbones (Tab. 4). The time is measured from the start of the diffusion sampling loop to the final decoded output. All models use identical sampling steps and inference settings for fairness. FeRA adds lightweight routing and frequency-aware operations, incurring only 4-8% overhead depending on backbone size, while remaining close to LoRA in efficiency and offering better generation quality and controllability.

## 6. Conclusion

In this paper, we presented FeRA, a parameter-efficient fine-tuning framework guided by the frequency-energy state of latent representations that utilizes a Frequency-Energy Indicator, Soft Router, and Consistency Loss to align adaptation with the natural frequency hierarchy. Extensive experiments demonstrate that FeRA achieves superior fidelity, alignment, and style customization with minimal parameter overhead. Our findings confirm that frequency-energy awareness offers a robust and interpretable direction for diffusion adaptation.

## Impact Statement

This paper presents work whose goal is to advance the field of Machine Learning. There are many potential societal consequences of our work, none which we feel must be specifically highlighted here.

## Acknowledgement

This work was supported by the National Natural Science Foundation of China under Grant No. 62320106007, and by NUS Grant A-0010106-00-00A-8004365-00-00 and A-8004410-01-00.

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

## A. Frequency-Energy Indicator Is Not a Proxy for Timestep

To validate the design motivation of FeRA, we investigate whether the Frequency-Energy Indicator (FEI) merely acts as a proxy for the diffusion timestep $t$. As shown in Fig. 8, we visualized the spectral evolution of 100 generation trajectories. **Consistent with standard diffusion notation where $t = 0$ represents the clean image and $t = T$ represents pure noise, the population mean exhibits a monotonic decrease, which means opposite direction to Fig. 1** This trend confirms the expected physical behavior: structural information degrades as noise intensity increases. However, this mean trend represents the limit of what static time embeddings can capture a global, uniform routing policy shared across all samples.

Crucially, the scatter plot reveals significant instance-specific variance around this mean. As annotated at $t = 500$, the high-frequency ratio spans a wide range (approx. $0.25$ to $0.60$), clearly distinguishing texture-rich instances from structurally simple ones at the exact same denoising stage. This empirical evidence confirms that FEI is not a proxy for time; unlike time-based methods, FeRA leverages this content-dependent variance to dynamically allocate expert capacity based on the actual complexity of the instance.

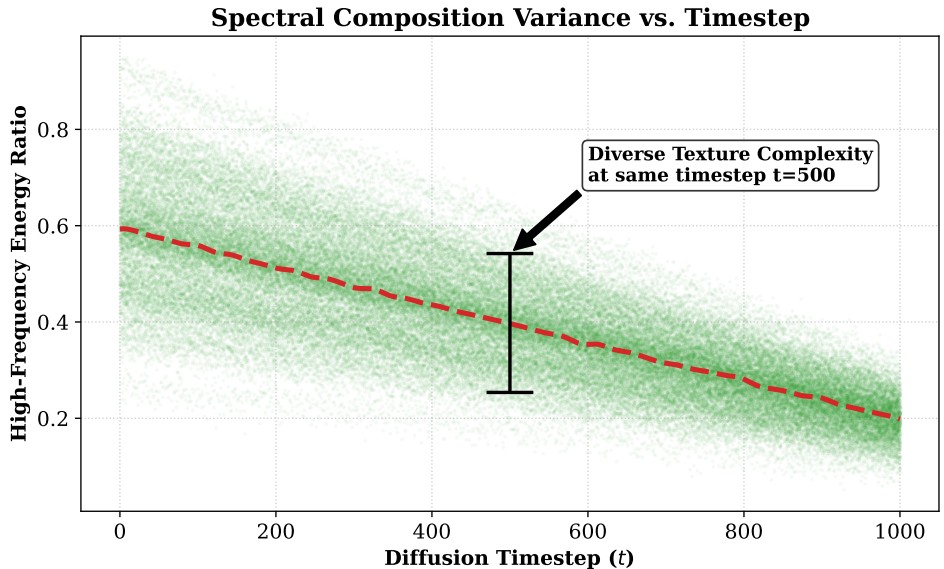

*Figure 8.* Instance-wise spectral variance.

## B. Theoretical Analysis of Frequency-Energy Routing

In this section, we provide a formal theoretical comparison between standard Time-Dependent Mixture-of-Experts (Time-MoE) and the proposed Frequency-Energy Routing (FeRA). Both of them use soft routing. We prove that FeRA implements a state-dependent control policy that strictly generalizes time-dependent policies, offering a superior optimization landscape by filtering updates on noise-dominated frequency bands.

### B.1. Routing Manifolds: Open-Loop vs. Closed-Loop

Let $\mathcal{Z} \subseteq \mathbb{R}^d$ denote the latent space of the diffusion model, and let $\{x_t\}_{t=0}^{T}$ represent the stochastic trajectory of the diffusion process.

**Definition 1 (Time-Dependent Routing).** A standard Time-MoE defines the routing weights $\mathbf{w} \in \Delta^{M-1}$ (where $\Delta$ is the simplex) as a function of the scalar timestep $t$:

$$\Phi_{time} : [0, T] \to \Delta^{M-1}, \quad \mathbf{w}_t = \text{Softmax}(\mathbf{W}_\tau \cdot \psi(t)) \tag{12}$$

where $\psi(t)$ is a fixed temporal embedding. This represents an *open-loop* control system where the expert selection is independent of the instantaneous state $x_t$. The decision boundary for Expert $k$ is defined by the level set $\{t \mid (\mathbf{w}_t)_k > \gamma\}$, which corresponds to fixed hyperplanes orthogonal to the time axis.

**Definition 2 (Frequency-Energy Routing).** FeRA defines the routing weights as a function of the spectral energy state of the latent $x_t$:

$$\Phi_{FeRA} : \mathcal{Z} \to \Delta^{M-1}, \quad \mathbf{w}_{freq} = \text{Softmax}\left(\frac{g_\phi(\mathcal{F}(x_t))}{\tau}\right) \tag{13}$$

where $\mathcal{F} : \mathcal{Z} \to \mathbb{R}^n$ is the Frequency-Energy Indicator (FEI) operator extracting band-wise energy, and $g_\phi$ is a learnable projection. This represents a *closed-loop* feedback system.

**Proposition 1 (Generalization Capability).** Since the expected spectral energy $\mathbb{E}[\mathcal{F}(x_t)]$ is monotonic with respect to $t$ (due to the diffusion schedule $\alpha_t$), there exists a mapping $h$ such that $\mathbb{E}[\Phi_{FeRA}(x_t)] \approx \Phi_{time}(t)$. However, since the instantaneous energy $\mathcal{F}(x_t)$ contains variance not captured by $t$ (i.e., entropy $H(\mathcal{F}(x_t)|t) > 0$), $\Phi_{FeRA}$ operates on a strictly richer manifold. $\Phi_{time}$ is a degenerate case of $\Phi_{FeRA}$ where the input is replaced by its population expectation.

## B.2. Gradient Orthogonality and Noise Suppression

We analyze the optimization behavior by examining the gradient applied to a specific expert adapter $\theta_k$ assigned to a high-frequency band. Consider the standard denoising objective $\mathcal{L} = \mathbb{E}_{x_0,\epsilon,t}[\|\epsilon - \epsilon_\theta(x_t, t)\|^2]$.

Let the residual error be decomposed into frequency components. The gradient update for expert $k$ is proportional to the router activation $w^{(k)}$:

$$\nabla_{\theta_k}\mathcal{L} \propto w^{(k)} \cdot \frac{\partial\mathcal{L}}{\partial\text{Output}} \tag{14}$$

**Theorem 1 (Gradient Noise in Time-MoE).** In Time-MoE, let $T_k$ be the time interval where expert $k$ is active (i.e., $w^{(k)}(t) \approx 1$). For a specific sample $x_t$, if the actual signal energy in band $k$ is zero (denoted as $S_k(x_t) = 0$), the neural network attempts to fit the pure noise component $\epsilon_k$. Since $w_{time}^{(k)}(t)$ is determined solely by $t$, if $t \in T_k$, then $w_{time}^{(k)} \gg 0$. Consequently:

$$\mathbb{E}[\|\nabla_{\theta_k}\mathcal{L}\| \mid S_k(x_t) = 0, t \in T_k] > 0 \tag{15}$$

This implies the expert $\theta_k$ receives non-zero gradient updates to fit pure Gaussian noise, leading to overfitting and optimization instability.

**Theorem 2 (Sparsity in FeRA).** In FeRA, the activation $w_{FeRA}^{(k)}$ is conditioned on the FEI, which approximates the signal-to-noise ratio. If a band $k$ lacks energy (i.e., $S_k(x_t) \approx 0$), the FEI vector reflects this sparsity, causing the router to suppress expert $k$:

$$S_k(x_t) \to 0 \implies \mathcal{F}(x_t)_k \to 0 \implies w_{FeRA}^{(k)} \to 0 \tag{16}$$

Therefore, the gradient magnitude is bounded by the signal presence:

$$\lim_{S_k \to 0} \|\nabla_{\theta_k}\mathcal{L}\|_{FeRA} = 0 \tag{17}$$

This property, which we term *Signal-Gradient Orthogonality*, ensures that FeRA experts are only updated when valid structural information is present, effectively performing dynamic curriculum learning unavailable to time-dependent routers.

## C. Generalization to Non-Natural Domains

A valid concern regarding frequency-aware methods is whether they rely excessively on natural image statistics (e.g., the $1/f$ power law), potentially limiting generalization to non-natural domains such as line art, cel-shaded anime, or abstract textures. We clarify that FeRA is designed to be domain-agnostic by utilizing the Frequency-Energy Indicator (FEI) as a sensing mechanism rather than a hard constraint. The FEI acts as a descriptive state indicator that measures the actual spectral composition of the latent features, regardless of whether that composition adheres to natural statistics.

The key to FeRA's robustness lies in its learnable routing network. During fine-tuning on a specific target domain, the router parameters are optimized to map the observed spectral signatures to the most effective expert combination. For instance, as illustrated in Fig. 9, domains like line art are inherently dominated by high-frequency edge information even at intermediate timesteps (e.g., $t = 500$), contrasting sharply with the low-frequency dominance of natural images. The router learns to interpret these high FEI values as essential structural features rather than noise, adaptively activating high-frequency experts. This contrasts with fixed priors that might suppress such signals.

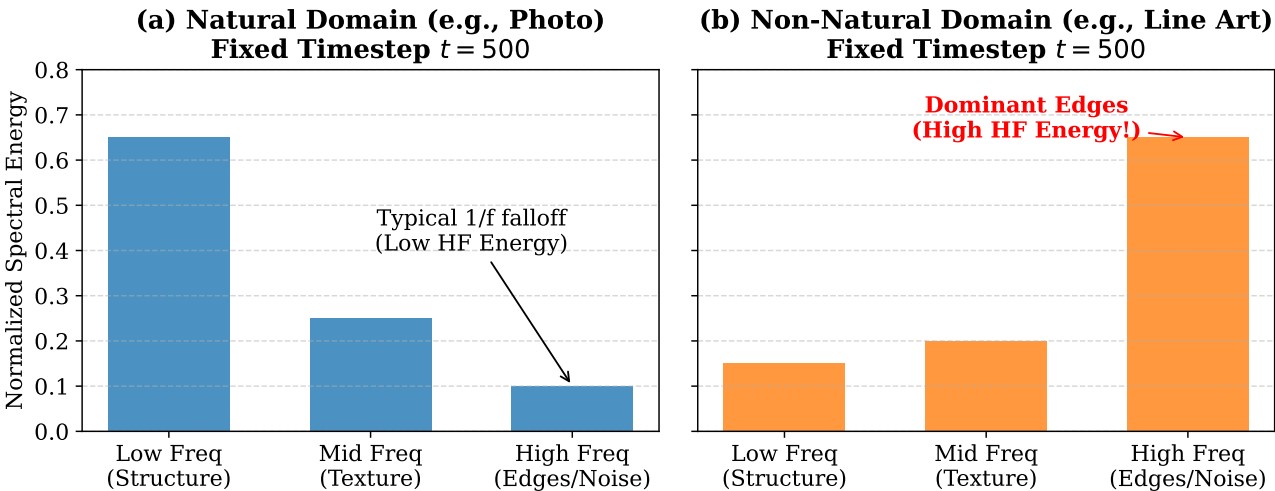

*Figure 9.* Spectral signature contrast at a fixed timestep ($t = 500$).

Consequently, rather than restricting generalization, frequency-awareness enhances the model's adaptability to out-of-distribution domains. While static, time-based baselines apply a uniform denoising policy blind to the domain's spectral shift, FeRA's instance-aware routing allows it to adjust its computational focus based on the specific complexity of the target domain, ensuring effective adaptation across both natural and artistic distributions.

## D. Experiment Setting

### D.1. MLLM-Judge Prompt

In experiment we evaluate stylistic fidelity(Style Score) using an MLLM-based style assessor. The model is prompted to judge the stylistic attributes of the generated images.For transparency and reproducibility, we include the exact prompt used in all evaluations below.

```
You are a strict T2I style judge.
Evaluate ONLY the STYLE qualities of the output image.
Use style cues from the PROMPT (and optional STYLE_TEXT / STYLE refs).
Ignore all semantic correctness.

Return a compact JSON with:  style_faithfulness, style_intensity,
palette_match, lighting_mood, texture_pattern, artifacts, overall_style.
Each score is in [0,10] (integer or one decimal).  No extra text.

Definitions:
- style_faithfulness:  reflection of style cues (medium, era, motifs).
- style_intensity:  stylization strength (not too weak or excessive).
- palette_match:  color palette and tone mapping.
- lighting_mood:  lighting style and ambience.
- texture_pattern:  local texture/strokes/pattern effects.
- artifacts:  fewer visual artifacts ⇒ higher score.
- overall_style = 0.30*style_faithfulness + 0.15*style_intensity
+ 0.20*palette_match + 0.15*lighting_mood
+ 0.20*texture_pattern.
```

### D.2. Training Setting

For clarity and reproducibility, we summarize the common training configuration used across all experiments in Table 5.

*Table 5.* Common experiment settings used throughout our study.

| Category | Setting |
| --- | --- |
| Base Model | frozen VAE and text encoder |
| LoRA Config | applied to attention modules (`to_q`, `to_k`, `to_v`, `to_out`) |
| Resolution | $512 \times 512$ (resize + center/random crop) |
| Batch Size | 16 per device |
| Training Steps | 5000 |
| Optimizer | AdamW, LR = $1 \times 10^{-4}$, weight decay = 0.01 |
| Warmup Steps | 500 |
| Scheduler | DDPM scheduler with standard noise schedule |
| Mixed Precision | fp16 |
| Grad Norm Clip | 1.0 |
| Inference | 30-step DDPM sampling |
| Hardware | NVIDIA H100 GPUs |

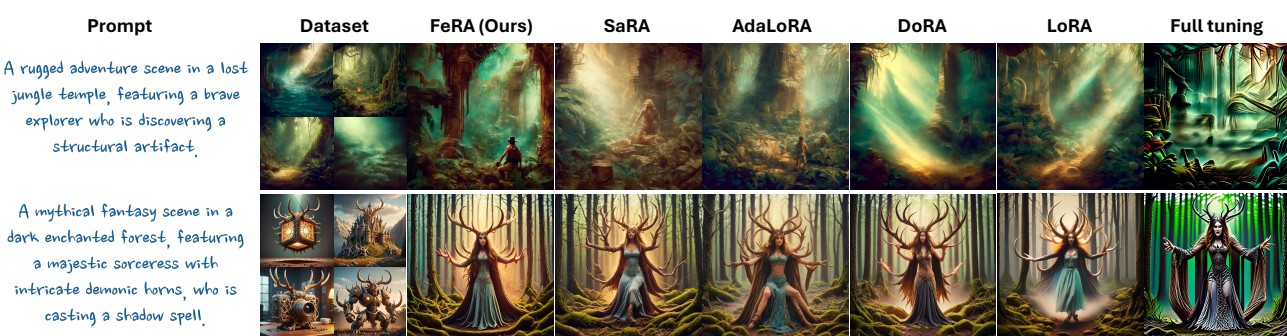

*Figure 10.* Comparison of the generated images between different PEFT methods in other datasets.

# E. Other Experiment Result

### E.1. Text-to-Image Style Adaptation

To further examine the generality of our training pipeline, we extend the text-to-image style adaptation experiments to multiple diffusion backbones with distinct latent resolutions, denoising trajectories, and text–image alignment capabilities. This broader evaluation helps disentangle whether the observed performance gains truly stem from our frequency–energy–guided fine-tuning strategy rather than from accidental synergy with a particular pretrained model. By comparing models that differ substantially in their architectural design and training dynamics, we can more reliably assess the stability and scalability of our method.

As summarized in Tab. 6, the performance trends remain consistent across Stable Diffusion 1.5 and Stable Diffusion XL. Despite the large gap in model capacity and latent-space structure, our approach produces clear improvements in perceptual quality, stylistic faithfulness, and controllability. At the same time, semantic alignment remains competitive, suggesting that the additional style expressiveness does not come at the cost of prompt consistency. These results indicate that our fine-tuning strategy transfers reliably across diffusion families and retains its benefits even as the backbone scale or architecture changes.

And we also show others qualitative result in other datasets or backbones in Fig. 10, 11 and 12.

### E.2. Image Customization

We further study instance-level image customization under DreamBooth fine-tuning on two backbones, Stable Diffusion 1.5 and Stable Diffusion 3.0. This setting evaluates how well different PEFT approaches preserve the identity of a specific object while retaining prompt alignment. Tab. 7 and 8 report CLIP-based image alignment (CLIP-I) and text alignment (CLIP-T).

Across both backbones, FeRA consistently achieves the best or second-best instance faithfulness (CLIP-I) while maintaining competitive or superior text alignment (CLIP-T) compared to LoRA, DoRA, AdaLoRA and SaRA. The trend is stable

*Table 6.* Comparison with different parameter-efficient fine-tuning methods on Stable Diffusion 1.5 and SDXL. **Orange bold** = best *Light orange italic* = second best. ***Notice:*** **the training parameter is same for fair comparison by controlling the rank of FeRA.**

| Backbone | Params | Method | Barbie | | | Cyberpunk | | | Expedition | | | Hornify | | | Elementfire | | |
|---|---|---|---|---|---|---|---|---|---|---|---|---|---|---|---|---|---|
| | | | CLIP ↑ | FID ↓ | Style ↑ | CLIP ↑ | FID ↓ | Style ↑ | CLIP ↑ | FID ↓ | Style ↑ | CLIP ↑ | FID ↓ | Style ↑ | CLIP ↑ | FID ↓ | Style ↑ |
| SD 1.5 | 5M | LoRA | **35.19** | 194.73 | 8.37 | 33.16 | *129.63* | 8.14 | 31.51 | 128.73 | 8.46 | 31.08 | 163.02 | 8.48 | *31.59* | 156.95 | 8.47 |
| | | DoRA | 35.12 | 193.90 | 8.36 | **33.20** | 131.51 | *8.19* | 31.53 | 129.13 | 8.47 | 31.15 | 164.24 | 8.50 | **31.62** | 158.59 | 8.49 |
| | | AdaLoRA | *35.12* | 194.30 | 8.38 | 33.12 | 131.51 | 8.17 | 31.53 | 129.00 | *8.50* | 31.15 | *162.81* | 8.49 | 31.55 | *156.62* | 8.51 |
| | | SaRA | 35.01 | *190.81* | *8.39* | *33.18* | 138.01 | 8.18 | *31.56* | *128.50* | 8.49 | *31.18* | 163.50 | *8.51* | 31.55 | 166.00 | *8.52* |
| | | **FeRA (Ours)** | 34.34 | **184.98** | **8.43** | 32.96 | **126.40** | **8.21** | **31.66** | **127.03** | **8.53** | **31.20** | **162.28** | **8.54** | 31.12 | **154.92** | **8.55** |
| | 20M | LoRA | **35.31** | 181.15 | 8.58 | 33.25 | 122.32 | *8.04* | 31.60 | 120.10 | 8.05 | 31.19 | 151.75 | 7.94 | **31.70** | 146.50 | 7.72 |
| | | DoRA | 35.25 | 180.32 | 8.01 | 33.22 | 122.32 | 7.59 | 31.63 | 120.12 | *8.13* | 31.25 | 152.12 | *8.34* | 31.64 | 155.32 | *8.27* |
| | | AdaLoRA | *35.25* | 180.70 | *8.60* | 33.22 | *122.30* | 7.83 | 31.63 | 119.95 | 7.58 | 31.25 | *151.45* | 8.31 | 31.62 | *146.11* | 8.17 |
| | | SaRA | 35.15 | *177.50* | 8.06 | *33.28* | 127.80 | 7.70 | *31.67* | *119.50* | 7.96 | *31.29* | 152.18 | 7.90 | *31.66* | 155.09 | 7.61 |
| | | **FeRA (Ours)** | 34.50 | **172.03** | **8.74** | **33.38** | **117.90** | **8.32** | **31.78** | **118.11** | **8.55** | **31.32** | **150.91** | **8.71** | 31.25 | **143.62** | **8.60** |
| | 50M | LoRA | **35.35** | 183.42 | 8.26 | *33.28* | 123.55 | 7.58 | 31.62 | 119.21 | 8.22 | 31.22 | 150.89 | *8.62* | **31.73** | *145.58* | 8.09 |
| | | DoRA | *35.29* | 182.63 | *8.63* | 33.25 | 123.52 | 7.43 | 31.65 | 119.22 | 8.44 | 31.28 | 151.63 | 7.69 | *31.69* | 154.72 | *8.21* |
| | | AdaLoRA | 35.29 | 182.92 | 8.16 | 33.25 | *123.35* | *8.19* | 31.65 | 119.05 | *8.52* | 31.28 | *150.47* | 8.24 | 31.69 | **145.54** | 7.83 |
| | | SaRA | 35.19 | *179.72* | 8.27 | **33.31** | 129.14 | 7.59 | *31.69* | *118.61* | *8.53* | *31.32* | 151.83 | 8.21 | 31.49 | 154.32 | 7.57 |
| | | **FeRA (Ours)** | 34.55 | **174.21** | **8.76** | 33.12 | 119.13 | **8.34** | **31.80** | **117.21** | **8.62** | **32.73** | 150.13 | **8.66** | 31.29 | 152.48 | **8.54** |
| | 860M | Full-Tuning | 34.48 | 176.50 | 8.70 | 33.05 | 121.20 | 8.30 | 31.75 | 119.00 | 8.58 | 31.30 | 151.80 | 8.63 | 31.20 | 154.00 | 8.51 |
| SDXL | 5M | LoRA | *35.88* | 188.41 | 8.53 | 33.45 | 157.92 | 8.38 | 31.94 | 141.52 | 8.60 | 32.45 | 155.82 | 8.66 | 32.32 | 173.73 | 8.67 |
| | | DoRA | 35.83 | 186.78 | 8.52 | 33.48 | 156.21 | 8.41 | 31.96 | 140.28 | 8.63 | 32.48 | 153.94 | 8.65 | 32.35 | 171.35 | 8.68 |
| | | AdaLoRA | 35.85 | 187.60 | *8.55* | 33.49 | 157.12 | 8.40 | 31.97 | 140.92 | 8.62 | 32.48 | 154.90 | 8.66 | 32.32 | 172.55 | *8.70* |
| | | SaRA | 35.76 | *183.25* | 8.54 | *33.51* | *152.69* | *8.43* | *31.98* | *138.15* | *8.64* | **32.51** | *151.07* | *8.67* | *32.38* | *168.22* | 8.69 |
| | | **FeRA (Ours)** | **37.86** | **178.67** | **8.57** | **33.61** | **148.88** | **8.44** | **32.08** | **134.69** | **8.67** | *32.49* | **147.29** | **8.70** | **32.48** | **163.51** | **8.72** |
| | 20M | LoRA | *35.99* | 175.23 | 8.36 | 33.57 | 146.84 | 8.74 | 32.07 | 131.42 | 8.44 | 32.58 | 144.90 | 8.28 | 32.45 | 156.42 | 8.05 |
| | | DoRA | 35.94 | 173.73 | 8.87 | 33.60 | 145.72 | 7.97 | 32.09 | 130.17 | 7.88 | 32.61 | 143.12 | *8.53* | 32.45 | 159.15 | *8.93* |
| | | AdaLoRA | 35.94 | 174.25 | 8.46 | 33.60 | 146.45 | *8.77* | 32.09 | 130.82 | *8.55* | 32.61 | 144.00 | 8.66 | 32.45 | 160.22 | 8.64 |
| | | SaRA | 35.87 | *170.44* | *8.88* | *33.63* | *141.93* | 7.91 | *32.11* | **128.26** | 8.22 | **32.64** | *140.23* | 8.30 | *32.51* | *156.25* | 8.27 |
| | | **FeRA (Ours)** | **36.97** | **166.61** | **8.94** | **34.73** | **138.45** | **8.82** | **32.21** | *128.29* | **8.68** | *32.62* | **136.38** | **8.92** | **33.70** | **152.11** | **8.95** |
| | 50M | LoRA | **36.01** | 176.41 | *9.09* | 33.59 | 147.19 | 8.45 | 32.08 | 130.18 | 8.23 | 32.59 | 144.24 | *9.11* | 32.46 | 160.58 | 8.23 |
| | | DoRA | 35.96 | 174.96 | 8.42 | 33.62 | 146.13 | 8.24 | 32.10 | 129.54 | 8.03 | 32.62 | 142.48 | 9.05 | 32.46 | 158.52 | 8.49 |
| | | AdaLoRA | 35.96 | 175.57 | 8.63 | 33.62 | 147.51 | *8.82* | 32.10 | 130.22 | 8.80 | 32.62 | 143.35 | 9.03 | 32.46 | 159.26 | *8.85* |
| | | SaRA | 35.89 | *171.66* | 8.89 | *33.65* | *143.52* | 8.57 | *32.12* | *127.66* | *8.24* | *32.65* | *139.61* | 8.98 | *32.52* | *155.56* | 8.72 |
| | | **FeRA (Ours)** | *35.99* | **167.33** | **9.12** | **33.75** | **139.51** | **9.05** | **33.22** | **124.21** | **8.99** | **33.63** | **136.11** | **9.23** | **33.62** | **151.46** | **8.86** |
| | 3.5B | Full-Tuning | 36.00 | 170.80 | 9.10 | 33.72 | 143.00 | 8.80 | 32.15 | 126.50 | 8.90 | 32.65 | 139.00 | 9.00 | 32.55 | 153.00 | 8.85 |

*Figure 11.* Comparison of the generated images between different PEFT methods(SD1.5).

on both the lighter SD 1.5 and the stronger SD 3.0 models, indicating that our framework provides robust benefits for DreamBooth-style image customization across different backbone capacities.

And we also show others qualitative result in other datasets in Fig. 13.

*Table 7.* Quantitative comparison between different PEFT methods on image customization (Stable Diffusion 1.5). **Red bold** = best, *orange italic* = second best.

| Methods | Dog | | Clock | | Backpack | | Toy Duck | | Teapot | |
|---|---|---|---|---|---|---|---|---|---|---|
| | CLIP-I ↑ | CLIP-T ↑ | CLIP-I ↑ | CLIP-T ↑ | CLIP-I ↑ | CLIP-T ↑ | CLIP-I ↑ | CLIP-T ↑ | CLIP-I ↑ | CLIP-T ↑ |
| Dreambooth + Full-tuning | 0.787 | 24.09 | 0.788 | 23.11 | 0.653 | 24.15 | 0.787 | 23.93 | 0.752 | 24.21 |
| Dreambooth + LoRA | 0.894 | 23.76 | 0.912 | 21.58 | *0.918* | 25.12 | 0.906 | 23.74 | 0.909 | 23.54 |
| Dreambooth + DoRA | *0.899* | 23.59 | *0.914* | 21.82 | 0.911 | *25.32* | *0.909* | 23.88 | *0.910* | 23.63 |
| Dreambooth + AdaLoRA | 0.898 | 23.74 | 0.911 | 21.82 | 0.917 | 25.39 | 0.907 | 23.93 | 0.905 | 23.67 |
| Dreambooth + SaRA | 0.788 | **25.98** | 0.884 | *23.59* | 0.887 | 25.35 | 0.882 | **25.51** | 0.868 | *24.99* |
| **Dreambooth + FeRA (Ours)** | **0.902** | *26.09* | **0.923** | **23.66** | **0.925** | **25.84** | **0.909** | *24.50* | **0.915** | **25.36** |

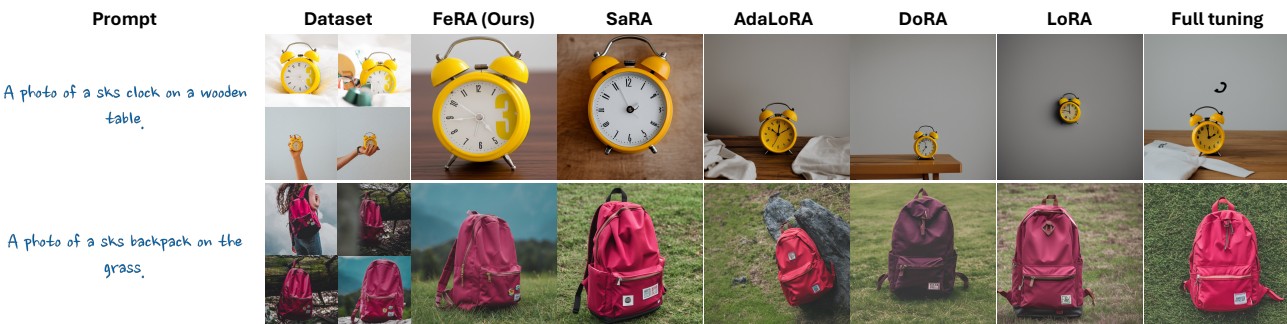

| Prompt | Dataset | FeRA (Ours) | SaRA | AdaLoRA | DoRA | LoRA | Full tuning |

A panther emerging from backstreets, its synthetic fur layered with glowing circuitry, droplets of acid rain scattering electric reflections.

A ferocious fire-born lynx sprinting through a burning ravine, its body stitched from flowing magma threads and bursting embers, each step leaving spirals of incandescent flames.

A determined jungle archaeologist navigating ancient overgrown corridors, uncovering a forgotten relic beneath moss-covered carvings as filtered sunlight.

*Figure 12.* Comparison of the generated images between different PEFT methods(SD3.0).

| Prompt | Dataset | FeRA (Ours) | SaRA | AdaLoRA | DoRA | LoRA | Full tuning |

A photo of a sks clock on a wooden table.

A photo of a sks backpack on the grass.

*Figure 13.* Comparison of the image customization between different PEFT methods.

*Table 8.* Quantitative comparison between different PEFT methods on image customization (Stable Diffusion 3.0). **Red bold** = best, *orange italic* = second best.

| Methods | Dog | | Clock | | Backpack | | Toy Duck | | Teapot | |
|---|---|---|---|---|---|---|---|---|---|---|
| | CLIP-I ↑ | CLIP-T ↑ | CLIP-I ↑ | CLIP-T ↑ | CLIP-I ↑ | CLIP-T ↑ | CLIP-I ↑ | CLIP-T ↑ | CLIP-I ↑ | CLIP-T ↑ |
| Dreambooth + Full-tuning | 0.798 | 24.55 | 0.799 | 23.55 | 0.664 | 24.49 | 0.800 | 24.45 | 0.760 | 24.52 |
| Dreambooth + LoRA | 0.905 | 24.04 | 0.923 | 22.11 | *0.927* | 25.63 | 0.915 | 24.20 | 0.916 | 23.98 |
| Dreambooth + DoRA | *0.907* | 24.11 | *0.925* | 22.18 | 0.924 | *25.71* | *0.917* | 24.28 | *0.918* | 24.05 |
| Dreambooth + AdaLoRA | 0.906 | 24.09 | 0.924 | 22.16 | 0.927 | 25.69 | 0.916 | 24.25 | 0.917 | 24.03 |
| Dreambooth + SaRA | 0.800 | **26.37** | 0.897 | *23.91* | 0.896 | 25.67 | 0.895 | **25.90** | 0.876 | *25.52* |
| **Dreambooth + FeRA (Ours)** | **0.910** | *26.35* | **0.930** | **24.03** | **0.935** | **25.75** | **0.920** | *25.00* | **0.923** | **25.72** |

## F. User Study (Human Evaluation)

While automated metrics (e.g., FID, CLIP Score, MLLM Score) provide quantitative insights, they may not fully align with human perceptual preferences, especially for fine-grained style adaptation tasks. To rigorously assess the perceptual quality of FeRA, we conducted a User Study comparing our method against the strongest baseline, LoRA.

**Experimental Setup.** We randomly selected $M = 30$ prompts from our test set, covering diverse styles. For each prompt, we generated image pairs using FeRA and LoRA with identical random seeds to ensure fairness. We invited $N = 20$ evaluators to perform a blind Two-Alternative Forced Choice test. For each pair, evaluators were shown the reference style image and the two generated results (in randomized order) and asked to select the better one based on two criteria:

- **Style Alignment:** Which image better captures the textures, brushstrokes, and color palette of the reference style?

- **Visual Fidelity:** Which image has better structural coherence and fewer artifacts?

**Results.** Figure 14 summarizes the voting results. FeRA significantly outperforms LoRA in human preference. Specifically, FeRA achieves a win rate of 68.5% in Style Alignment and 62.0% in Visual Fidelity.

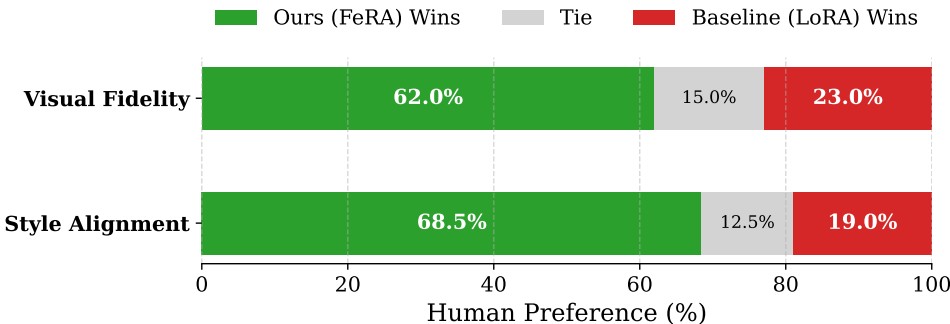

*Figure 14.* User study.

