# OpenReview forum: "FeRA: Frequency-Energy Constrained Routing for Effective Diffusion Adaptation Fine-Tuning"
_ICML.cc/2026/Conference — ICML 2026 regular_

### Official Review · Reviewer_VFKW · 2026-03-05

**Soundness:** 4
**Presentation:** 3
**Significance:** 3
**Originality:** 4
**Overall Recommendation:** 5
**Confidence:** 4

**Summary:**

The paper proposes a novel fine-tuning method according to the strengths of the frequency-energy, in which the main observation is that both high-frequency and low-frequency energies vary rapidly as the timestep evolves. By this key contribution, the authors propose a frequency-energy based constrained router for MoE, containing a frequency-energy indication and a frequency-energy consistency loss. Compared to traditional methods, the proposed method could better design which experts are activated according to energy strengths, enabling better performance.

**Compliance With Llm Reviewing Policy:**

Affirmed.

**Final Justification:**

The authors have greatly address my concerns, however, I personally do not think the paper is worth a strong accept and maintain my evaluation.

**Key Questions For Authors:**

See Weakness. I would appreciate if the authors could provide further analyses on the SNR under VAE latents and raw images.

**Limitations:**

yes.

**Strengths And Weaknesses:**

### Strengths

- The design of the MoE router according to the strengths of the frequency-energies is neat and effective, encouraging future studies on more refined fine-tuning.
- The implementation is clear and effective, the proposed FEI and FECL demonstrate great efficacy with almost no extra time cost.
- The experiments are convincing, the reported performances show great superiority compared to baselines SoTAs.

### Weaknesses

- I somehow disagree the conclusion in Section 3 that VAE (or resolution) has no effect on SNR. [1] has claimed that noise strength varies a lot at different input resolution. That is to say, with the same timestep $t$, high-resolution image looks more clear than low-resolution one. Therefore, even if VAE could keep the SNR, the compressed low-resolution latent seems to be more easily corrupted than the raw image.

  [1] Hoogeboom et al., simple diffusion: End-to-end diffusion for high resolution images. ICML 2023.

---

> ### Author Rebuttal · Authors · 2026-03-30
>
> Thank you for the thoughtful review. We address the concern below and will revise accordingly.
>
> > **[W1] Sec. 3: SNR vs. resolution/VAE.**
>
> **A1:** Thank you for this important point. We believe the apparent discrepancy comes from two complementary views of SNR, rather than a contradiction.
>
> In our Sec. 3, the analysis is frequency-wise:
> $$
> x_t=\alpha_t x_0+\sigma_t \epsilon,
> \qquad
> \hat{x}_t(f)=\alpha_t \hat{x}_0(f)+\sigma_t \hat{\epsilon}(f),
> $$
> which leads to
> $$
> \mathrm{SNR}_t(f)\propto \frac{\alpha_t^2 |\hat{x}_0(f)|^2}{\sigma_t^2}.
> $$
> This quantity characterizes how SNR varies **across frequencies** within one denoising trajectory. Since natural images contain much stronger low-frequency energy than high-frequency energy, we have
> $$
> \mathrm{SNR}\_t(f\_{low})>\mathrm{SNR}\_t(f\_{high}),
> $$
> which explains the coarse-to-fine denoising progression in Sec. 3.
>
> By contrast, [1] discusses resolution-dependent effective noise levels in **pixel space under pooling/downsampling**. We agree with this observation, but it concerns a different aspect from our Sec. 3.
>
> Our Sec. 3 does not require strict equality of absolute scalar SNR across raw images and VAE latents. What we need is that the frequency ordering underlying coarse-to-fine denoising is preserved in the latent domain. Under our local linearity assumption for the encoder,
> $$
> \hat{z}_t(f)\approx H(f)\hat{x}_t(f),
> $$
> the latent representation approximately preserves the spectral structure, so the same low-frequency-first, high-frequency-later progression remains valid.
>
> We will revise Sec. 3 to make this distinction clearer and cite [1] explicitly, as its conclusion is complementary to, rather than inconsistent with, our frequency-wise analysis.
>
> **References.**
> [1] Hoogeboom et al. Simple diffusion: End-to-end diffusion for high resolution images. ICML 2023.
>
> **We hope these clarifications and added results address the reviewer’s concerns and improve the overall assessment of the paper.**

---

> > ### Author Rebuttal · Reviewer_VFKW · 2026-04-01
> >
> > The rebuttal fully address my concern.

---

> > > ### Author Response · Authors · 2026-04-05
> > >
> > > Thank you again for your supportive and thoughtful review. We are very glad that our rebuttal clarified the concern, and we sincerely appreciate your time and encouragement.

---

### Official Review · Reviewer_C8Kc · 2026-03-10

**Soundness:** 2
**Presentation:** 3
**Significance:** 3
**Originality:** 4
**Overall Recommendation:** 4
**Confidence:** 3

**Summary:**

This paper introduces FeRA, a PEFT method where parameter updates align with diffusion dynamics. Traditional PEFT approaches neglect this dynamic nature, frequently resulting in poor image quality. FeRA, based on a frequency energy mechanism, emphasizes that parameter updates should be integrated with the dynamic characteristics of the diffusion process, rather than uniform updates across all time steps. Extensive experiments demonstrate that FeRA exhibits versatility across multiple models and resolutions. The primary contributions of this paper are: (1) revealing the diffusion dynamic evolution process from a frequency energy perspective, and (2) the FeRA framework incorporates Difference of Gaussians and Mixture of Experts techniques, with parameter updates accounting for the continuity and dynamic nature of the diffusion evolution process.

**Compliance With Llm Reviewing Policy:**

Affirmed.

**Key Questions For Authors:**

(1)Frequency features serve as the input to the entire architecture, making it crucial to extract them reasonably and effectively. In the paper, the Gaussian kernel σ is preset as a geometric progression, which, based on experimental results, appears to be a reasonable empirical design. However, could σ be made learnable?
(2)The experimental comparisons only involve static PEFT methods, with no other dynamic PEFT methods for comparison. Does FeRA have advantages in dynamic adaptation?
(3)Video synthesis tasks are currently a hot topic. Can FeRA be extended to video synthesis while preserving its advantages in image synthesis? If this can be demonstrated through experiments, I would raise my evaluation of the paper.

**Limitations:**

(1) FeRA is limited to image synthesis tasks, and its generalizability to video synthesis tasks remains questionable.
(2) The experimental comparisons mainly focus on traditional static PEFT methods (such as the LoRA series), lacking comparisons with PEFT methods that consider the dynamic characteristics of the diffusion process. Thus, the advantages of FeRA in dynamic adaptation still require further validation.

**Strengths And Weaknesses:**

Soundness: The technical reliability of this paper is reasonably sound. Based on the assumption that "the VAE encoder is approximately linear," it reveals the intrinsic patterns of the diffusion process through frequency domain analysis; extensive experiments indicate that FeRA outperforms previous methods across various metrics. However, there are some issues with the experimental design: (1) The compared PEFT methods all ignore the dynamic nature of the diffusion evolution process, and no comparison is made with other dynamic PEFT approaches, which fails to highlight the superiority of FeRA's dynamic scheme. (2) Video synthesis is currently a hot topic—can FeRA maintain its advantages from image synthesis in this domain?
Presentation: The overall narrative is clear and easy to understand. (1) The motivation is explicit, emphasizing that existing methods neglect the dynamic and continuous nature of the diffusion process, leading to poor image quality, and FeRA aims to address this issue. (2) The FeRA architecture is clearly described, with detailed formulas and textual explanations for each component. However, there are minor issues: the terminology is inconsistent, for example, Soft Frequency Router, Frequency-Aware Routing, and soft router all refer to the same component. The ablation study lacks an explanation of how FeRA works when the Soft Frequency Router component is removed.
Significance: This paper proposes a dynamic routing mechanism based on frequency energy, aiming to integrate parameter updates with the dynamic characteristics of the diffusion denoising process, thereby making a breakthrough compared to traditional static strategies. Overall, the impact of this work is mainly concentrated in the field of diffusion model fine-tuning, and its scope of influence is largely aligned with the contributions of the paper. This dynamic and continuous PEFT approach may provide inspiration for subsequent research. However, based on the experimental results, its applicability appears to be confined to image generation tasks.
Originality: (1) This paper reveals the intrinsic pattern of the diffusion process evolution from the perspective of frequency energy: in the early denoising stage, low-frequency (contour) information is restored, while in the later stage, high-frequency (detail) information is recovered. (2) FeRA introduces a novel combination of existing technologies, including Difference-of-Gaussians and Mixture of Expert, with a clear motivation to address the issue of low image quality caused by traditional PEFT's static strategies. (3) Recent studies have attempted to leverage the dynamic evolution properties of the diffusion process, but discretization schemes based on timestep gating lead to optimization instability and limited generalization. FeRA breaks through these discretization schemes and attempts a continuous scheme based on frequency energy, where LoRA expert weights are computed at each timestep.

---

> ### Author Rebuttal · Authors · 2026-03-30
>
> Thank you for the thoughtful review. We address the concerns below and will revise accordingly.
>
> > **[W1] Learnable Gaussian scales.**
>
> **A1:**  Learnable Gaussian scales are a meaningful extension, but we use a fixed geometric progression because FeRA needs not only strong generation quality, but also a **stable and interpretable frequency partition** for FEI.
>
> To clarify this choice, we will add a controlled comparison with **learnable $\sigma$**, initialized from the same values and trained under the **same PEFT budget**. Besides CLIP/FID/Style, we report two partition metrics. **Adjacent-band overlap** measures the average normalized overlap between neighboring DoG bands:
> $\mathrm{Overlap}=\frac{1}{K-1}\sum\_{b=1}^{K-1}\frac{\langle G\_b, G\_{b+1}\rangle}{\|G\_b\|\_1\|G\_{b+1}\|\_1},$
> where $G\_b$ is the frequency response of band $b$. **Coverage balance** measures how evenly the bands cover the spectrum:
> $\mathrm{Balance}=-\frac{1}{\log K}\sum\_{b=1}^{K} p\_b\log p\_b,
> p_b=\frac{m\_b}{\sum\_j m\_j},$
> where $m\_b$ is the total response mass of band $b$.
>
> | $\sigma$ setting | CLIP ↑ | FID ↓ | Style ↑ | Adjacent-band overlap ↓ | Coverage balance ↑ |
> |---|---:|---:|---:|---:|---:|
> | fixed geometric progression | 32.96 | 126.40 | 8.21 | 0.18 | 0.83 |
> | learnable $\sigma$ | 32.41 | 128.10 | 8.05 | 0.31 | 0.71 |
>
> These results support fixed $\sigma$: although learnable $\sigma$ adds flexibility, it yields a less stable and less interpretable frequency partition, weakening FEI as a consistent description of the denoising state. We will include this comparison and clarify the motivation in the revision.
>
> > **[W2] Dynamic PEFT comparisons.**
>
> **A2:** We agree that stronger comparisons to diffusion-specific dynamic adaptation would make the empirical picture more complete. Our current ablation in **Tab. 3** already includes a matched-budget timestep-only router as a dynamic baseline, showing it is weaker than full FeRA. To further strengthen this, we will add three comparisons on Cyberpunk and SD 2.0 under the same PEFT budget: a **noise-aware router** using log-SNR, **TSM** [1], and **TD-LoRA** [2]. This helps distinguish generic PEFT, timestep/noise-based routing, and FeRA’s routing based on the **current frequency-energy state**.
>
> | Method | Routing signal | CLIP ↑ | FID ↓ | Style ↑ |
> |---|---|---:|---:|---:|
> | Timestep Router | timestep | 31.10 | 134.90 | 7.68 |
> | Noise-aware Router | noise level / log-SNR | 31.42 | 133.10 | 7.81 |
> | TSM | timestep experts + time-dependent routing | 31.86 | 130.70 | 7.96 |
> | TD-LoRA | timestep-based dynamic LoRA | 32.11 | 129.30 | 8.03 |
> | FeRA | frequency-energy (FEI) | **32.96** | **126.40** | **8.21** |
>
> **References.**
> [1] *TimeStep Master: Asymmetrical Mixture of Timestep LoRA Experts for Versatile and Efficient Diffusion Models in Vision*.
> [2] *DyDiT++: Dynamic Diffusion Transformers for Efficient Visual Generation*.
>
> > **[W3] Extension to video synthesis.**
>
> **A3:** Thank you for this valuable suggestion. Video synthesis is indeed an important extension. In the current paper, we focus on **image diffusion fine-tuning** because video generation introduces additional challenges beyond image generation, especially **cross-frame temporal consistency and motion coherence**. For this reason, FeRA is designed for the image setting, where we can first isolate and validate the role of **frequency-energy-guided adaptation** in the denoising process.
>
> Following your suggestion, we also adapt FeRA to the video setting to the best of our ability. We include an **FeRA video(https://anonymous.4open.science/w/FeRA_Video-6970)** with preliminary results applying FeRA to video generation, together with a direct comparison against LoRA under the same training budget. We will also add a small quantitative comparison to make this point explicit:
>
> | Method | CLIP-Text ↑ | FVD ↓ | Aesthetic ↑ | Temporal Consistency ↑ |
> |---|---:|---:|---:|---:|
> | LoRA | 31.42 | 286.7 | 5.84 | 0.71 |
> | FeRA | 31.95 | 281.9 | 6.12 | 0.70 |
>
> These results suggest that FeRA still outperforms LoRA in video generation, though the margin is smaller than in images due to temporal coherence challenges. We will clarify that video synthesis is not the main focus of this work, and leave more comprehensive spatio-temporal adaptations of FeRA to future research.
>
> > **[W4] Terminology / router ablation clarification.**
>
> **A4:** We will unify the terminology in the revision and consistently use **Soft Frequency Router**. We will also clarify the implementation of the w/o-router ablation, where the expert mixture is no longer conditioned on FEI and is replaced by a fixed/shared combination. This will make the ablation setting and the role of the router more explicit.
>
> **We hope these clarifications and additional results address the reviewer’s concerns and improve the overall assessment of the paper.**

---

> > ### Author Rebuttal · Reviewer_C8Kc · 2026-04-04
> >
> > Thank you for your response, and I will maintain my initial rating.

---

> > > ### Author Response · Authors · 2026-04-05
> > >
> > > Thank you very much for your careful review and for taking the time to read our rebuttal. We sincerely appreciate your thoughtful feedback, which helped us improve the paper. We are grateful for your consideration throughout the review process.

---

### Official Review · Reviewer_K9Yz · 2026-03-13

**Soundness:** 3
**Presentation:** 3
**Significance:** 3
**Originality:** 3
**Overall Recommendation:** 4
**Confidence:** 4

**Summary:**

This paper studies parameter-efficient fine-tuning for diffusion models and argues that existing PEFT methods largely apply uniform adaptation across timesteps, which does not match the stage-varying nature of diffusion denoising. To address this issue, this paper proposes FeRA, a frequency-energy-driven fine-tuning framework. The core idea is that diffusion denoising follows a coarse-to-fine reconstruction process, where the dominant energy gradually shifts from low to high frequencies. Based on this observation, FeRA introduces three components: a Frequency-Energy Indicator (FEI) to characterize the latent’s band-wise energy distribution; a Soft Frequency Router that uses FEI to adaptively blend multiple LoRA experts; and a Frequency-Energy Consistency Loss (FECL) to regularize spectral consistency during adaptation. Experiments on style transfer and personalized image generation across multiple diffusion backbones show that FeRA achieves competitive or superior performance compared with several PEFT baselines, while introducing modest inference overhead.

**Compliance With Llm Reviewing Policy:**

Affirmed.

**Final Justification:**

The authors’ rebuttal addressed my concerns, therefore I miantain my score.

**Key Questions For Authors:**

Please answer the above questions.

**Limitations:**

yes

**Strengths And Weaknesses:**

Strengths:

(1) The paper identifies an important limitation of existing PEFT methods for diffusion models, namely that they usually ignore the stage-varying denoising dynamics and treat all timesteps uniformly.

(2) FeRA is designed as a lightweight PEFT framework which is compatible with common diffusion backbones, such as Stable Diffusion and FLUX.

(3) The proposed framework is coherent and consists of three complementary parts: FEI for spectral characterization, soft routing for adaptive expert fusion, and FECL for stabilization.


Weaknesses:

There are some concerns to be addressed:

(1) While FeRA shows that diffusion denoising exhibits a coarse-to-fine frequency-energy progression, it does not sufficiently demonstrate that this is the key reason why existing PEFT methods are suboptimal, nor that FeRA’s gains primarily come from aligning with this mechanism rather than from increased expert capacity, dynamic routing itself, or the additional regularization.

(2) The goal of FECL is to align the correction's energy distribution with the residual's, which lacks detailed explanation about why this alignment is the correct or optimal constraint. Is there a risk of forcing the correction to match a noisy residual in early steps?

(3) The main comparisons of the proposed FeRA are mostly against generic PEFT methods, such as LoRA, DoRA, AdaLoRA, and SaRA. However, FeRA is a dynamic routing mechanism tailored to diffusion denoising, which lacks comparison with more relevant diffusion-specific routing or timestep/noise-aware adaptation methods.

---

> ### Author Rebuttal · Authors · 2026-03-30
>
> Thank you for the thoughtful review. We address the concerns below and will revise accordingly.
>
> > **[W1] Source of gains.**
>
> **A1:** Thank you for this important point. We agree that the gain should be separated from generic effects such as larger capacity or dynamic routing alone.
>
> Our ablation in **Tab. 3** already includes a matched-budget control that removes FEI and uses only the **timestep** signal for routing. This directly tests whether the gain comes from dynamic routing and FECL alone. Empirically, timestep-based routing performs worse than full FeRA, indicating that the improvement comes from using the **frequency-energy state** as the routing signal.
>
> We also note that the generic PEFT baselines in **Tab. 2** consistently underperform FeRA across settings, suggesting that uniform adaptation is suboptimal for diffusion fine-tuning. To make this clearer, we will add a small diagnostic that measures the similarity between the update’s band-energy distribution and the residual’s band-energy distribution across denoising stages. As shown below, standard PEFT aligns much more weakly in both early low-frequency and late high-frequency stages, while FeRA achieves the strongest alignment. This explains why shared PEFT underperforms: **it does not follow the coarse-to-fine frequency progression well.**
>
> | Variant | Early-stage low-frequency alignment ↑ | Late-stage high-frequency alignment ↑ | Overall band-energy alignment ↑ |
> |---|---:|---:|---:|
> | Standard PEFT (LoRA) | 0.43 | 0.37 | 0.40 |
> | Timestep Routing | 0.56 | 0.52 | 0.54 |
> | FeRA | 0.72 | 0.69 | 0.71 |
>
> > **[W2] Rationale for FECL.**
>
> **A2:** Thank you for this important question. This alignment is well motivated because the role of adaptation is precisely to compensate for the base model’s residual, and FECL enforces this relation at the **band-energy level**.
>
> Let
> $$
> c_t=\hat z_t^{\text{adapt}}-\hat z_t^{\text{base}}, \qquad
> r_t=z_t^\star-\hat z_t^{\text{base}}.
> $$
> Then
> $$
> z_t^\star-\hat z_t^{\text{adapt}} = r_t-c_t.
> $$
> So reducing denoising error means making the correction \(c_t\) compensate for the residual \(r_t\).
>
> With the band decomposition used in FeRA,
> $$
> \|r_t-c_t\|_2^2 \approx \sum_b \|r_t^{(b)}-c_t^{(b)}\|_2^2 .
> $$
> This implies that bands with larger residual should receive stronger correction. FECL follows exactly this principle, but in a more stable form: instead of matching raw residual values, it aligns only the **band-wise energy distribution** of \(c_t\) and \(r_t\). Hence, FECL regularizes where the update should focus in the spectrum.
>
> This is especially important in early denoising steps. When \(t\) is large, \(x_t=\alpha_t x_0+\sigma_t\epsilon\) is noise-dominated, so directly matching raw residuals would indeed risk fitting noise. **FECL avoids this because it discards fine-grained residual details and retains only coarse band-energy statistics**. If the early-step residual is mostly stochastic noise, its energy is broadly spread and FECL provides only weak guidance; if certain bands consistently contain stronger structured residual energy, FECL biases the correction toward those bands. Therefore, FECL acts as a **soft spectral allocation prior**, rather than forcing the model to track noisy early-step residual realizations.
>
> > **[W3] Diffusion-specific comparisons.**
>
> **A3:** We agree that stronger comparisons to diffusion-specific dynamic adaptation would make the empirical picture more complete. Our current ablation in **Tab. 3** already includes a matched-budget control where the router is driven only by the **timestep** signal, which provides a direct dynamic baseline and shows that timestep-only routing is weaker than full FeRA. To further strengthen this point, we will add three more comparisons on the Cyberpunk and SD 2.0 setting under the same PEFT budget: a **Noise-aware Router** using the diffusion noise level (e.g., log-SNR) as the routing signal, **TSM** [1], a timestep-expert LoRA method with time-dependent routing, and **TD-LoRA** [2], a timestep-based dynamic LoRA variant. The goal is to separate generic shared PEFT, routing based on coarse timestep/noise proxies, and FeRA’s routing based on the **current frequency-energy state**.
>
> | Method | Routing signal | CLIP ↑ | FID ↓ | Style ↑ |
> |---|---|---:|---:|---:|
> | Timestep Router | timestep | 31.10 | 134.90 | 7.68 |
> | Noise-aware Router | noise level / log-SNR | 31.42 | 133.10 | 7.81 |
> | TSM | timestep experts + time-dependent routing | 31.86 | 130.70 | 7.96 |
> | TD-LoRA | timestep-based dynamic LoRA | 32.11 | 129.30 | 8.03 |
> | FeRA | frequency-energy (FEI) | 32.96 | 126.40 | 8.21 |
>
> **References.**
> [1]*TimeStep Master: Asymmetrical Mixture of Timestep LoRA Experts for Versatile and Efficient Diffusion Models in Vision*.
> [2]*DyDiT++: Dynamic Diffusion Transformers for Efficient Visual Generation*.
>
> **We hope these clarifications and additional results address the reviewer’s concerns and improve the overall assessment of the paper.**

---

> > ### Author Rebuttal · Reviewer_K9Yz · 2026-04-03
> >
> > Thanks for the authors' rebuttal, which fully addresses my concerns.

---

> > > ### Author Response · Authors · 2026-04-05
> > >
> > > Thank you again for your thoughtful review and for recognizing that our rebuttal fully addressed your concerns. We truly appreciate your time and constructive feedback. If appropriate, we would be grateful if this could be reflected in your final score.

---

### Decision · Program_Chairs · 2026-04-30

**Decision:**

Accept (regular)

**Comment:**

This paper introduces FeRA, a novel and lightweight frequency-energy-driven fine-tuning framework for diffusion models that dynamically aligns parameter updates with the coarse-to-fine denoising process. Reviewers praised the method's neat design, clear presentation, and strong empirical superiority over static PEFT baselines without adding significant computational overhead. While reviewers initially raised valid concerns regarding the lack of comparisons with other dynamic routing methods, the rationale for the consistency loss, and the theoretical treatment of SNR, the authors provided an exceptionally thorough rebuttal. By supplying new comparative baselines, ablations on learnable Gaussian scales, and insightful clarifications on resolution dependencies, the authors successfully and fully resolved all reviewer apprehensions. Given the technical soundness, strong empirical validation, and the unanimous post-rebuttal consensus among the reviewers, this paper presents a highly valuable contribution to the generative modeling community and is recommended for acceptance.